

# A stand-alone calibration approach for attitude-based multi-copter wind measurement systems

Matteo Bramati[1], Martin Schön[1], Daniel Schulz[1], Vasileios Savvakis[1], Jens Bange[1], and Andreas Platis[1]

[1]Eberhard Karls Universität Tübingen, Geo- und Umweltforschungszentrum, 72076 Tübingen

**Correspondence:** Matteo Bramati (matteo.bramati@uni-tuebingen.de)

**Abstract.** The determination of the wind profile of the lower atmospheric boundary layer is an important aspect of meteorology and wind energy science. A suitable tool to capture the wind profile is the usage of small unmanned aircraft systems (UAS). This study describes an easily repeatable type of calibration process in order to obtain an estimation of the horizontal wind vector using a rotary-wing UAS in hovering conditions. This procedure works without using wind tunnels or meteorological

masts: it requires only the data from the flight control unit and a particular set of calibration flights. A modified DJI S900 hexacopter has been used for this study. The UAS body has been encased in a styrofoam sphere, leaving only the rotors and the landing gear outside, in order to grant a higher level of isotropy with respect to the incoming wind flow. A model based on the characterization of the UAS drag coefficient is proposed for the estimation of the relative horizontal wind vector. Validation flights have been performed at the German Weather Service MOL-RAO observatory in Falkenberg, Brandenburg. By hovering

aside of a 99 m high meteorological mast with ultrasonic anemometers, it was possible to prove the wind prediction capability and assess the accuracy of the model. The analysis of the power spectral density highlights how the system resolves atmospheric eddies up to 0.1 Hz frequency. The overall root mean square error is less than 0.7 ms$^{-1}$ for the wind speed while less than 8 deg for the wind direction.

## 1 Introduction

Information about the properties of the wind field in the lower atmospheric boundary layer is essential for meteorological applications like the development of weather models or industrial applications like the design of wind power plants (Chioncel et al., 2011; Platis et al., 2020, 2021). Common in situ wind field measurement devices are cup or ultrasonic anemometers together with wind vanes. These instruments are widely used for weather forecasting, but they need to be mounted on a mast providing only one-point measurement. Remote sensing methods like LiDAR and SoDAR are less spatially restricted but

exhibit a lower resolution and a higher requirement for maintenance. All of these methods are limited by varying applicability, measurement errors, need for maintenance and cost.

Rotary-wing unmanned aircraft systems (UAS) - also called, in this text, multi-copters - represent an alternative for measuring the horizontal wind-vector. These systems can take off and land vertically and are able to hover at a fixed point in space. Their use is comparatively inexpensive, and they can fly autonomously without prior knowledge of the wind field (Waslander

and Wang, 2009). The measurement strategy is highly flexible and can be planned on short notice before the take-off: ver-





tical or horizontal profiles can be performed as well as grid mapping at specific locations. Several integrated sensors can be mounted on a multi-copter providing insightful information about temperature, pressure, humidity, $CO_2$ (Segales et al., 2020). Also, more complex systems could be installed, like sonic anemometers, in order to sample the three-dimensional wind speed directly (Shimura et al., 2018). Even though flight endurance is still an issue for these systems, automatic landing procedures

and self-recharging stations could automate the operability soon.

The dependency of the UAS attitude on the incoming wind provides the basis for estimating the wind-vector itself: it is a common approach for horizontal wind estimation. Neumann and Bartholmai (2015) introduced this method and performed the calibration of the UAS in a wind tunnel obtaining an estimation model with a root mean square error (RMSE) of 0.6 ms$^{-1}$. Palomaki et al. (2017) compared an attitude based wind estimation to a sonic anemometer mounted on a multi-copter, obtaining

comparable statistics. Crowe et al. (2020) applied machine learning algorithms to model the relation between wind speed and multi-copter attitude. Wetz et al. (2021) recently performed UAS fleet flights measuring the vertical profile of the wind-vector and compared the results with LiDAR data, finding an excellent agreement and even higher temporal/spatial resolution.

Usually, the UAS calibration requires the usage of wind tunnels or masts equipped with anemometers. We present a more feasible approach that consists in calibrating the wind-attitude relation performing specific missions with the system flying at

different constant ground speed (GS) under low atmospheric wind conditions (Brosy et al., 2017). This procedure allows for in situ calibration of the system in the natural atmospheric environment; it grants a complete mapping of the UAS attitude up to the maximum flight speed and can be easily repeated for different multi-copters. The possible presence of non-negligible atmospheric wind in the moment of the calibration results in the ground speed being different from the true air speed (TAS). Even in this adverse scenario, we show that data can still be used to obtain a calibration function after correcting them with a

dedicated post-processing algorithm.

The wind measurement model developed in this study is based on the calibration of a multi-copter using its drag coefficient as a function of the UAS attitude. Wetz et al. (2021) introduced this method by assuming a linear behaviour for this aerodynamic parameter while using the UAS in a wind-vane configuration (always facing the incoming wind).

Our study modified the multi-copter external shape by enclosing all the electronic components into a styrofoam sphere. The

resulting increased isotropy allows us to avoid the wind-vane mode: this is essential since our system would be sensibly slow in turning into the wind direction due to its size. The proposed model is explained in detail by analysing the behaviour of the drag coefficient versus the modified-shape UAS attitude. This approach is then compared to the model mapping directly the system attitude into the wind velocity, introduced by Neumann and Bartholmai (2015). Further advantages of the new spherical-shape configuration coupled with the drag coefficient model are highlighted throughout the text.

**Section 2** describes the UAS used for this study. The theory of wind estimation is also outlined. The steps that connect the full rigid-body dynamics equations and the final explicit relation between the horizontal wind and the multi-copter attitude are explained by highlighting all the hypotheses made.

**Section 3** explains the strategy used for collecting calibration data. The post-processing of these data is depicted, and the drag coefficient model is introduced together with the Neumann and Bartholmai (2015) model.

**Section 4** outlines the procedure used to assess the quality of the two models. Plots of time series and spectra are reported.





**Table 1.** Relevant specifications of the UAS used in this study.

| | |
|---|---|
| **frame** | DJI Spreading Wings S900 |
| **flight control unit** | PixHawk 2.1 Cube Orange |
| **GNSS** | Here3 GPS |
| **rotor-rotor distance** | 900 mm |
| **Sphere Diameter** | 500 mm |
| **multi-copter weight** | 3.8 kg |
| **batteries weight** | 3.0 kg |
| **sphere weight** | 0.5 kg |
| **total weight** | 7.3 kg |

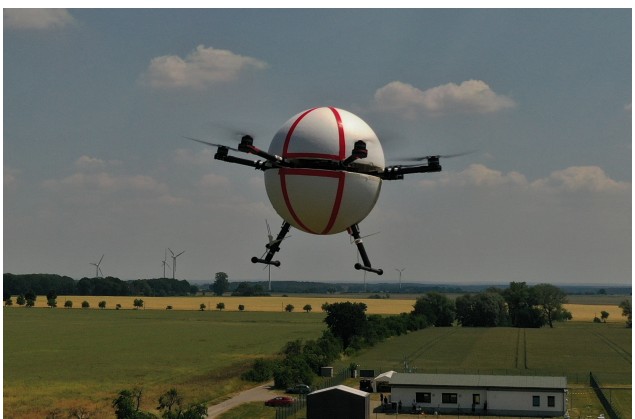

**Figure 1.** Modified DJI S900 hexacopter during a mission.

In **Sect. 5** an extensive analysis of the results is carried out, including the quality of the wind-vector estimation and the influence of several parameters.

Finally in **Sect. 6**, conclusions of this study are drawn and possible future developments are outlined.

## 2 Materials and methods

### 2.1 Unmanned aerial system


The system used for this study is a rotary-wing UAS, precisely the frame of the hexacopter DJI Spreading Wings S900. The system is controlled by a PixHawk 2.1 Cube Orange autopilot, configured with the open-source ArduCopter firmware. A Here3 GPS antenna is used as a GNSS by the flight control unit. The autopilot creates a log file containing the flight parameters of the system, as computed by its extended Kálmán filter, during every mission and logged at 10 Hz frequency.





The body of the hexacopter has been encased in a styrofoam sphere (Fig. 1). This shape grants increased isotropy with respect to the horizontal wind and, at the same time, provides a partial shelter to the electronics from external environmental conditions. The usefulness of this encasing will be discussed more in detail in Sect. 5.

    The power is provided by a pair of 6 cells Lithium Polymer batteries with a capacity of 12 Ah each. The system weighs around 7.3 kg including the batteries: the maximum flight time is approximately 24 minutes. In this configuration the hexa-
copter proved to fly up to a peak GS of 19 ms$^{-1}$.

    Mission Planner is the software used on the ground station. This open-source program allows to create missions with specific commands, simulate them, and subsequently load them on the system memory. The same interface is used during flights to monitor the orientation, the battery level, the altitude and several other crucial parameters.

### 2.2   Wind estimation theory

#### 2.2.1   Reference frames

Several reference frames can be used to describe the orientation of an airborne system. In this study only two of them are needed:

- **Inertial reference frame or North-East-Down** ($i_1$, $i_2$, $i_3$):
  This reference frame has its origin in the center of gravity of the multi-copter: $i_1$ points towards north, $i_2$ points towards
east and $i_3 = i_1 \times i_2$ will consequently point towards the center of the Earth.

- **Body reference frame** ($b_1$, $b_2$, $b_3$):
  This reference frame shares the same origin with the NED. However, its axes move together with the vehicle: $b_1$ points towards the front of the vehicle, $b_2$ points towards the right of the vehicle and $b_3 = b_1 \times b_2$ will consequently be orthogonal to the plane defined by the first two.

It is possible to express variables such as position or velocity of the system with respect to one frame or the other: a change of reference frame is performed applying three planar rotations defined by the so-called Euler angles. The rotation sequence is roll($\phi$)-pitch($\theta$)-yaw($\psi$) in order to switch from Body to NED while vice versa to switch from NED to Body. This operation is performed mathematically by using a rotation matrix:

$$\mathbf{R}_{N \to B} = \mathbf{R}_\phi(\phi)\mathbf{R}_\theta(\theta)\mathbf{R}_\psi(\psi) \tag{1}$$
$$\mathbf{R}_{B \to N} = \mathbf{R}_\psi(\psi)\mathbf{R}_\theta(\theta)\mathbf{R}_\phi(\phi) \tag{2}$$

The full expression of the rotation matrix can be found in Beard and McLain (2012).

#### 2.2.2   Tilt angle $\Gamma$

The tilt angle is defined as the angle between the NED vertical vector ($i_3$) and the Body reference frame vertical vector ($b_3$). A simplified representation of the airborne system and the tilt angle is shown in Fig. 2. Making use of the rotation matrix $\boldsymbol{R}_{B \to N}$



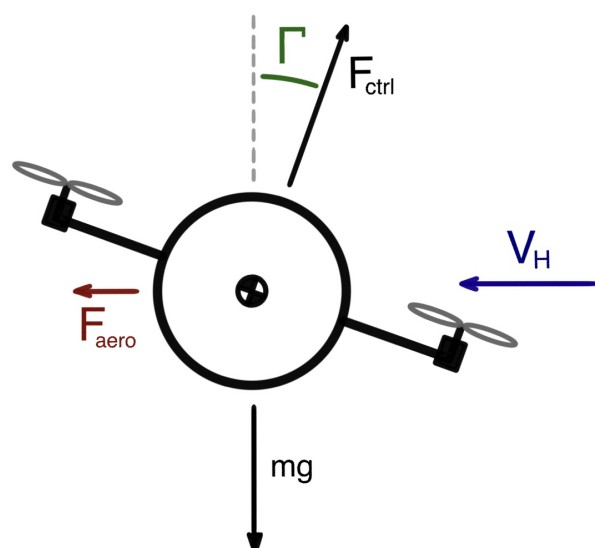

**Figure 2.** Schematic representation of the multi-copter attitude when flying at constant speed or hovering at a fixed position under the influence of a horizontal atmospheric wind. The multi-copter tilts ($\Gamma$) towards the wind direction, using its $F_{\mathrm{ctrl}}$ (thrust) in order to balance the aerodynamic force ($F_{\mathrm{aero}}$).

it is possible to obtain an expression for the tilt angle as a function of the pitch and roll angle only. Indeed, by expressing the $\boldsymbol{b}_3$ vector in the NED:

$$\boldsymbol{b}_3^{\mathrm{N}} = \mathbf{R}_{\mathrm{B}\to\mathrm{N}} \cdot \boldsymbol{b}_3^{\mathrm{B}} \tag{3}$$

$$\boldsymbol{b}_3^{\mathrm{N}} = \begin{bmatrix} \mathrm{c}(\phi)\mathrm{s}(\theta)\mathrm{c}(\psi) + \mathrm{s}(\phi)\mathrm{s}(\psi) \\ \mathrm{c}(\phi)\mathrm{s}(\theta)\mathrm{s}(\psi) - \mathrm{s}(\phi)\mathrm{c}(\psi) \\ \mathrm{c}(\phi)\mathrm{c}(\theta) \end{bmatrix} \overset{\psi=0}{=} \begin{bmatrix} \mathrm{c}(\phi)\mathrm{s}(\theta) \\ -\mathrm{s}(\phi) \\ \mathrm{c}(\phi)\mathrm{c}(\theta) \end{bmatrix} \tag{4}$$

where $\mathrm{c}() = \cos()$ and $\mathrm{s}() = \sin()$.

Here the yaw angle ($\psi$) is set to zero since, intuitively, a system's rotation around its vertical axis is not going to affect the tilt value. Thus the tilt angle can be computed using the three components of the $\boldsymbol{b}_3^{\mathrm{N}}$ vector:

$$\Gamma = \arctan\left[\frac{\sqrt{\mathrm{c}(\phi)^2\mathrm{s}(\theta)^2 + \mathrm{s}(\phi)^2}}{\mathrm{c}(\phi)\mathrm{c}(\theta)}\right] \tag{5}$$

### 2.2.3   Dynamics equations

The equations describing the dynamics of a rigid airborne system moving in still air are:

$$m\dot{\boldsymbol{v}} = m\boldsymbol{v} \times \boldsymbol{\omega} + \boldsymbol{F}_{\mathrm{aero}}(\boldsymbol{v}) + mg\boldsymbol{i}_3 - F_{\mathrm{ctrl}}\boldsymbol{b}_3 \tag{6}$$

$$I\dot{\boldsymbol{\omega}} = I\boldsymbol{\omega} \times \boldsymbol{\omega} + \boldsymbol{M}_{\mathrm{aero}}(\boldsymbol{\omega}, \boldsymbol{v}) + \boldsymbol{M}_{\mathrm{ctrl}} \tag{7}$$



where $m$ is the mass of the vehicle, $\boldsymbol{v}$ is its three component velocity vector, $\boldsymbol{\omega}$ its three component rotation rate vector, $\boldsymbol{F}_{\mathrm{aero}}$ and $\boldsymbol{M}_{\mathrm{aero}}$ are the aerodynamic forces and moments developing during flight while $F_{\mathrm{ctrl}}$ and $\boldsymbol{M}_{\mathrm{ctrl}}$ are the control forces and moments used by the autopilot in order to achieve a specific mission (Gonzalez-Rocha et al., 2017).

Considering steady ($\dot{\boldsymbol{v}} = 0$ and $\dot{\boldsymbol{\omega}} = 0$) equilibrium conditions ($\boldsymbol{v} = \boldsymbol{v}_{\mathrm{eq}}$ and $\boldsymbol{\omega} = 0$) the equations become:

$$0 = \boldsymbol{F}_{\mathrm{aero}}\left(\boldsymbol{v}_{\mathrm{eq}}\right) + mg\boldsymbol{i}_3 - F_{\mathrm{ctrl}}\boldsymbol{b}_3 \tag{8}$$

$$0 = \boldsymbol{M}_{\mathrm{aero}}\left(0, \boldsymbol{v}_{\mathrm{eq}}\right) + \boldsymbol{M}_{\mathrm{ctrl}} \tag{9}$$

Equation (8) has three components along the three NED directions. Assuming a condition of horizontal flight ($w_{\mathrm{eq}} = 0$) and that no vertical aerodynamic forces develop if the previous condition is matched, Eq. (8) can be written as:

$$F_{\mathrm{aero}}^{\mathrm{x}}\left(u_{\mathrm{eq}}\right) - F_{\mathrm{ctrl}}\,\mathrm{c}(\phi)\mathrm{s}(\theta) = 0 \tag{10}$$

$$F_{\mathrm{aero}}^{\mathrm{y}}\left(v_{\mathrm{eq}}\right) + F_{\mathrm{ctrl}}\,\mathrm{s}(\phi) = 0 \tag{11}$$

$$mg - F_{\mathrm{ctrl}}\,\mathrm{c}(\phi)\mathrm{c}(\theta) = 0 \tag{12}$$

Thus the total aerodynamic force acting on the vehicle will be:

$$F_{\mathrm{aero}} = \sqrt{(F_{\mathrm{aero}}^{\mathrm{x}})^2 + (F_{\mathrm{aero}}^{\mathrm{y}})^2} \tag{13}$$

$$= F_{\mathrm{ctrl}}\sqrt{\mathrm{c}(\phi)^2\mathrm{s}(\theta)^2 + \mathrm{s}(\phi)^2} \tag{14}$$

Computing the ratio between Eq. (14) and Eq. (12):

$$\frac{F_{\mathrm{aero}}}{mg} = \frac{F_{\mathrm{ctrl}}\sqrt{\mathrm{c}(\phi)^2\mathrm{s}(\theta)^2 + \mathrm{s}(\phi)^2}}{F_{\mathrm{ctrl}}\,\mathrm{c}(\phi)\mathrm{c}(\theta)} = \tan(\Gamma) \tag{15}$$

$$F_{\mathrm{aero}} = mg \cdot \tan(\Gamma) \tag{16}$$

Under the assumption made throughout the calculations it is possible to obtain an equation for the aerodynamic forces as a
function of the mass of the multi-copter and its orientation in space.

### 2.2.4   Aerodynamic Forces

A solid body with a relative velocity with respect to a surrounding fluid experiences a distribution of forces over its surface. This distribution is usually simplified by taking the equivalent components parallel and perpendicular to the relative velocity, and, depending on the reference point, an equivalent aerodynamic moment (Anderson, 2011). The parallel component is directed
opposite to the relative velocity and is often called aerodynamic drag. Under the assumption made in order to obtain Eq. (16), the drag is the only aerodynamic force acting on the system.

The absolute value of this component is characterised using the Rayleigh equation (Anderson, 2011):

$$D = \frac{1}{2}\rho V^2 A C_{\mathrm{D}}(\mathrm{Ma}, \mathrm{Re}) = F_{\mathrm{aero}} \tag{17}$$





where $\rho$ is the fluid density, $V$ is the relative velocity between the body and the fluid, $A$ is the projection of the body shape
exposed to the flow (cross-section area) and $C_D$ is a coefficient called drag coefficient. This last is a function of Mach number,
Reynolds number and the geometry of the body.

First of all, under the assumptions made in Sect. 2.2.3 - precisely the equilibrium hypothesis ($\boldsymbol{v} = \boldsymbol{v}_{\mathrm{eq}}$ and $\boldsymbol{\omega} = 0$) - the
velocity in Eq. (17) represents only the horizontal velocity. Moreover, the particular shape of our UAS offers an increased
level of axial symmetry in the horizontal plane with respect to a fixed-wing aircraft or other common multi-copters due to the
styrofoam sphere. Thus it is a valid assumption to consider that physical properties such as $A$ and $C_D$ do not depend on the
direction of motion. The dependence of the drag coefficient on the Mach number can also be neglected since the expected
range of velocities (up to $20 \ \mathrm{ms^{-1}}$) is considerably lower than the speed of sound.

### 2.2.5   Wind Speed Estimation

Combining equations 16 and 17 leads to a non-linear relation between $V$ and the tilt angle:

$$\frac{\rho V^2 A C_D}{2mg} = \tan(\Gamma) \tag{18}$$

$A$ and $C_D$ are not necessarily constant when the tilt (or the velocity) varies. Being the drag coefficient a function of the
Reynolds number, it can not be considered constant a priori when flying at different speeds. The relation $A(\Gamma)$ considers all
the multi-copter physical components outside the styrofoam sphere (rotors and landing gear) that modify the cross-section area
when the system is tilting.

Since the mapping of the $A(\Gamma)$ would be a complex procedure, it is useful to define a new extended drag coefficient:

$$C_A(\Gamma) = \frac{A(\Gamma)}{A_0} \cdot C_D(\Gamma) \tag{19}$$

Where $A_0$ is an arbitrary reference area used to keep the coefficient dimensionless: in this case the styrofoam sphere cross-
section area is used. $C_D$ is now a function of the tilt angle since there is a unique relation between $\Gamma$ and $V$.

It is now possible to isolate $V$ from Eq. (18):

$$V = \sqrt{\frac{2mg \cdot \tan(\Gamma)}{\rho \cdot A_0 \cdot C_A(\Gamma)}} = \sqrt{\frac{\tan(\Gamma)}{K \cdot C_A(\Gamma)}} \tag{20}$$

with:

$$K = \frac{\rho A_0}{2mg} \tag{21}$$

So far, $V$ was defined as the multi-copter horizontal velocity relative to the surrounding air; however, all the equations are
likewise valid when the system is hovering in the presence of a horizontal wind component. This condition can be easily
achieved by selecting the autopilot PosHold mode so that the UAS will maintain its position as computed by the onboard
extended Kálmán filter.





There are then two options in order to get an estimation of the horizontal wind. One is to characterize the relation between tilt angle and velocity directly (Neumann and Bartholmai, 2015), while the other consists in characterizing the behaviour of the extended drag coefficient against the tilt angle and then applying Eq. (20).

Both options require a UAS calibration procedure that is usually performed in wind tunnels or using meteorological masts with higher accuracy reference sensors (Neumann and Bartholmai, 2015; Palomaki et al., 2017). The method presented in this study does not depend on the availability of elaborate equipment: it consists of performing specific flights in the real environment in order to gather orientation data during constant speed sections (Brosy et al., 2017).

An issue of this calibration procedure is that the multi-copter can only be programmed to keep a specific ground speed while

flying, but the velocity $V$ that appears in Eq. (17) and (20) is the true air speed. The only case in which there is no difference between GS and TAS is when the atmospheric wind is zero. In other words, the presence of any non-zero wind during the flights makes the GS different from the TAS. It is then necessary to plan the calibration flights properly and adopt a systematic post-processing procedure that corrects the tilt angle taking into account the possible presence of atmospheric wind.

### 2.2.6 Wind direction estimation

The calculation of the wind direction ($\Lambda$) is straight forward. Since it is assumed that the multi-copter tilts in the direction of the wind, it is enough to calculate the projection of the $\boldsymbol{b}_3^{\mathrm{B}}$ vector on the horizontal plane of the NED reference frame. In this case the yaw angle ($\psi$) of the system needs to be accounted:

$$\Lambda = \arctan\left[\frac{c(\phi)s(\theta)}{-s(\phi)}\right] + \psi + 180° \tag{22}$$

The last addition is necessary since there is a 180° difference, by definition, from the projected $\boldsymbol{b}_3^{\mathrm{B}}$ and the meteorological wind

direction. After limiting $\Lambda$ to be in the range between 0 and 360°, Eq. (22) provides the horizontal wind direction from north, positive in clock-wise direction.

## 3 Calibration

The calibration flights were performed at the airfield located in Poltringen, Baden-Württemberg, Germany (UTC+1), on the 23rd of February, 2021.

The goal of the calibration flights is to collect data in order to map the multi-copter behaviour at different TAS. These data will help to obtain a direct relation between TAS and tilt angle and between the $C_{\mathrm{A}}$ and the tilt angle.

To put the conditions in the measurement area in Poltringen on the measurement day into a broader context, we use ERA5 reanalysis (Hersbach et al., 2018) data from Baden-Württemberg. The 23rd of February was a day with low wind speeds, low cloud cover and warm temperatures for February, between 17°C at the solar maximum and 10°C towards the evening. The

wind situation during the first two flights (Table 2 and Fig. 4, 14:00-15:00 UTC) remained constant, in the region around Poltringen with low surface wind speeds around 1 ms$^{-1}$ and an easterly wind direction.





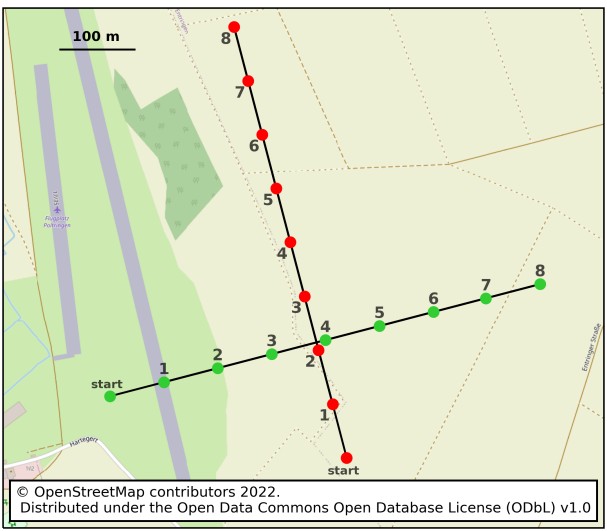

**Figure 3.** Example of the calibration flights missions (8-1 ms$^{-1}$). The green and red points correspond to limit of each section for a specific ground speed. Lower velocities require less distance since the multi-copter reaches a steady condition faster.

Between 15:00 UTC and 16:00 UTC, and thus between flight 2 and flight 3, the wind conditions changed. The wind direction changed from east to south-east at 16:00 and finally to south at 17:00. At the same time, the surface wind speed increased to around 2 ms$^{-1}$ at 16:00 UTC and to 2-3 ms$^{-1}$ at 17:00 UTC (Fig. 4, 16:00-17:00 UTC).

The UAS performed its mission at 50 m altitude on a straight line towards a waypoint and eventually back to the starting point. This basic flight pattern was carried out for different GS from 1 to 14 ms$^{-1}$ (safety limit). After completing this first set of flights, the procedure was repeated for a direction perpendicular to the first one to gather more data for the model calibration. The missions were planned in such a way to have an equal amount of data points for each GS, resulting in shorter distances for the lower velocities while larger distances for the higher ones (Fig. 3).

Flying forward and backward on a straight line is the key for detecting the influence of any non-zero atmospheric wind. Indeed if any wind is present at the moment of these flights, it will disturb the UAS with the same magnitude but in an opposite way. Thus it becomes easy to detect this influence and correct the recorded values in the post-processing so that the final calibration function will involve the TAS and not the GS (see Sect. 3.1 and 3.2).

     The total number of flights has been divided into 4 missions due to the endurance of the batteries. The mission sequence is 210   reported in Table 2.

     Figure 5 shows an example of the behaviour of the GS for the first mission.

     Atmospheric parameters were measured at ground level in order to calculate the air density.





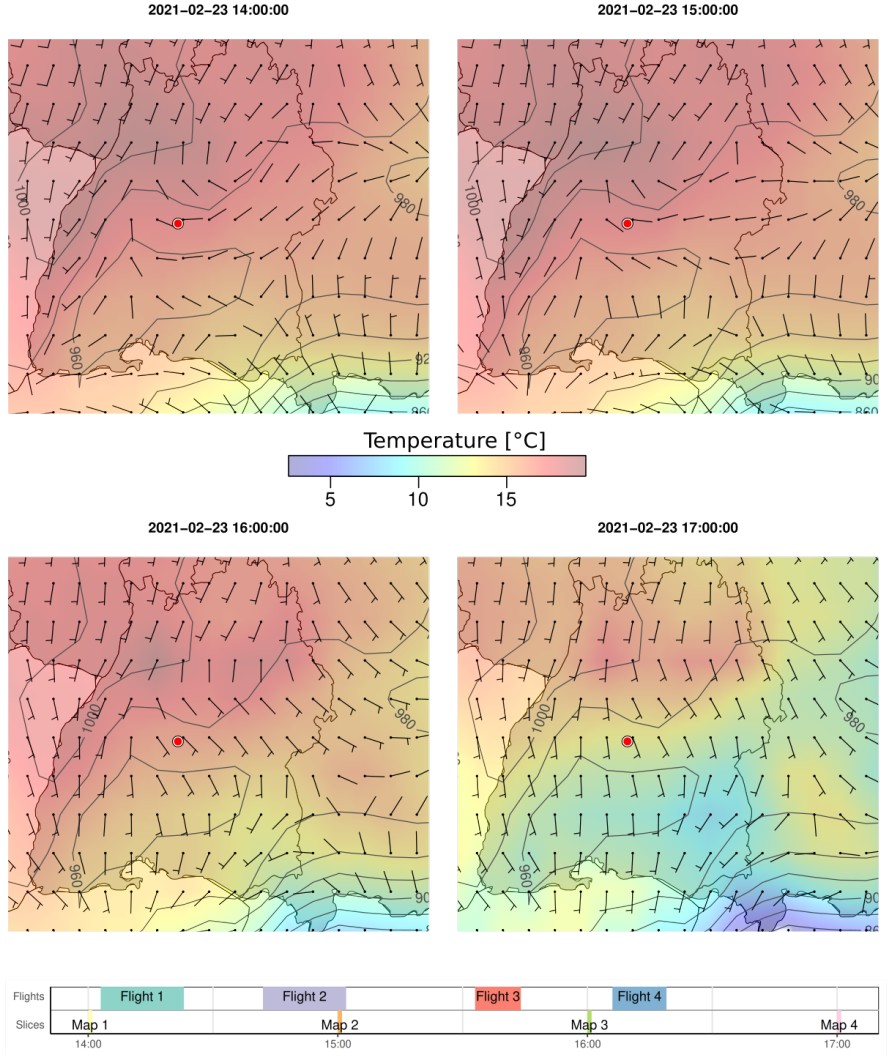

**Figure 4.** Maps describing the atmospheric conditions (surface wind speed, direction, temperature and pressure) at Poltringen airfield - the red dot on the maps - on the day of the calibration flights, February 23, 2021. The four plots describe the evolution of these parameters from 14:00 UTC (upper-left) to 17:00 UTC (lower-right). The ERA5 reanalysis was used to generate the plots. In the lower part a timeline shows when the four flights have been performed with respect to the four weather maps provided.

### 3.1 Data analysis

First, the data collected by the autopilot have been filtered around the desired GS with a band of +-0.05 ms$^{-1}$. The band is
uniform for all the tested velocities (red segments in Fig. 5). Subsequently, the tilt angle and the extended drag coefficient have





**Table 2.** Missions sequence. In Fig. 3, the E-W direction is represented by the green dots while N-S by the red dots. The surface wind speed (SWS) and direction (SWD) are obtained using ERA5 reanalysis.

| No. | Flight | GS [ms$^{-1}$] | Start [UTC] | End [UTC] | SWS [ms$^{-1}$] | SWD |
|---|---|---|---|---|---|---|
| **1** | E-W | 8-1 | 14:03 | 14:23 | $\leq 1$ | E |
| **2** | N-S | 8-1 | 14:42 | 15:02 | $\leq 1$ | E |
| **3** | E-W | 14-10 | 15:33 | 15:44 | $\approx 2$ | S-E |
| **4** | N-S | 14-10 | 16:06 | 16:19 | $\approx 2$ | S-E |

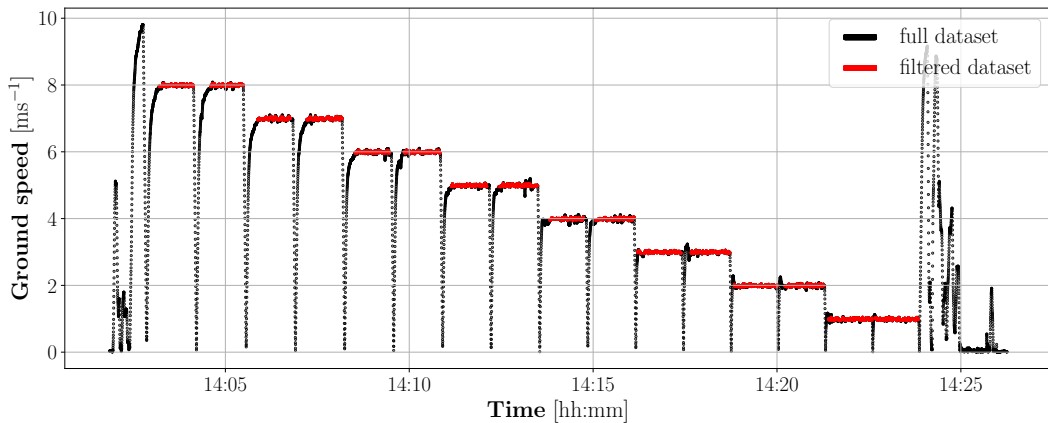

**Figure 5.** Example of the ground speed data recorded during mission No. 1 (E-W direction, see Table 2). The red part represents the filtered data used for the calibration. For each ground speed two data sets were obtained from flying first towards the waypoint and then back.

been computed starting from the Euler angles by applying Eq. (5) and the inverse of Eq. (20):

$$C_{\mathrm{A}} = \frac{\tan(\Gamma)}{K \cdot V^2} \tag{23}$$

At this point the only way to solve Eq. (23) is to assume undisturbed wind conditions (TAS = GS) for the moment and use the multi-copter $V_{\mathrm{GS}}$ instead of the TAS ($V$).

$$C_{\mathrm{A}} = \frac{\tan(\Gamma)}{K \cdot V_{\mathrm{GS}}^2} \tag{24}$$

Figure 6 shows the $\Gamma(V_{\mathrm{GS}})$ data and the $C_{\mathrm{A}}|_{\mathrm{GS}}(\Gamma)$ data obtained for all the tested GSs for one flight direction.

The data recorded during the first mission (8-1 ms$^{-1}$) started at 14:03 UTC do not show any sensible difference between forward and backward part, and in general they are very well overlapped. In contrast, data gathered during the mission No. 3 mapping higher GSs, started at 15:33 UTC show two very well separated clouds of points: this is clear evidence of the presence

of wind influencing the calibration. First, the multi-copter has to tilt more because it is facing the wind (head-wind); on its way





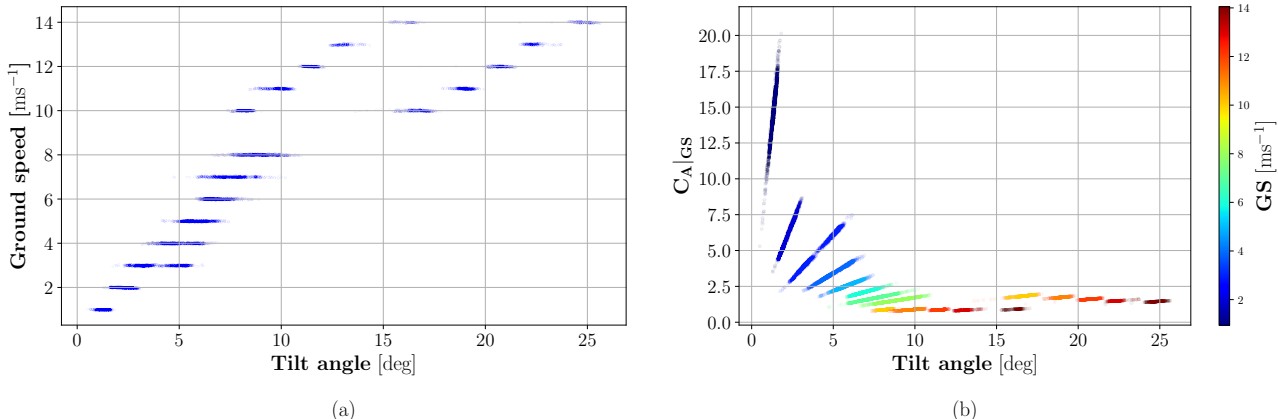

<p style="text-align:center">(a)</p>

<p style="text-align:center">(b)</p>

**Figure 6. a)** Tilt angles values recorded along the E-W direction (mission No. 1 and 3) for the ground speeds from from 1 to 8 ms$^{-1}$ and 10 to 14 ms$^{-1}$, respectively. **b)** Extended drag coefficient values obtained using Eq. (24) along the same direction plotted against the respective tilt angle values. The colour represents the different ground speeds.

back, it has to tilt less since it is pushed from behind (tail-wind). This behaviour (not shown here) also repeats for the second flight direction: it shows that the wind velocity picked up between the second and the third mission.

It should be noted how the variance of the tilt angle, along the single forward or backward leg, does not worsen when the external wind field is present, even though the data filtering has not been directly performed on the angle itself, but on the GS.

This variance, which can be seen in Fig. 6a for each GS, reflects directly into the shape of Fig. 6b. In order to understand this influence, it is useful to remember that for values of the tilt angle $\leq 20°$, the tangent of an angle approximates the angle itself. The range of values of the tilt angles, in this case, allows to apply this approximation, so the equation represented in Fig. 6b is:

$$C_\mathrm{A}|_\mathrm{GS} \approx \frac{\Gamma}{K \cdot V_\mathrm{GS}^2} \,, \tag{25}$$

which is a straight line passing for the origin of the axes for every ground speed $V_\mathrm{GS}$.

A variance of the tilt angle at a specific GS will end up in obtaining a segment along a line whose characteristic slope is determined by the GS itself (assuming constant mass and gravitational field). This can be seen in Fig. 6b where 8 lines are present for the velocities from 1 to 8 ms$^{-1}$ while for the flights from 10 to 14 ms$^{-1}$ we have two well separated segments for each velocity, due to the presence of wind. It is worth noticing that the dependence of the extended drag coefficient $C_\mathrm{A}$ on the

tilt angle $\Gamma$ is much stronger for smaller velocities (around 10 for 1 ms$^{-1}$) compared to higher velocities (less than 1 at 14 ms$^{-1}$).

An eventual deviation of the GS from the mean value would cause a change in the characteristic slope of each line. Due to the relatively narrow filtering of the data applied at the beginning of the analysis, this phenomenon is almost totally negligible. However, since the filtering band was uniform over all the different GSs, once more, this effect would have a higher impact at lower velocities as a uniform filtering band results in a higher relative error at lower GSs.





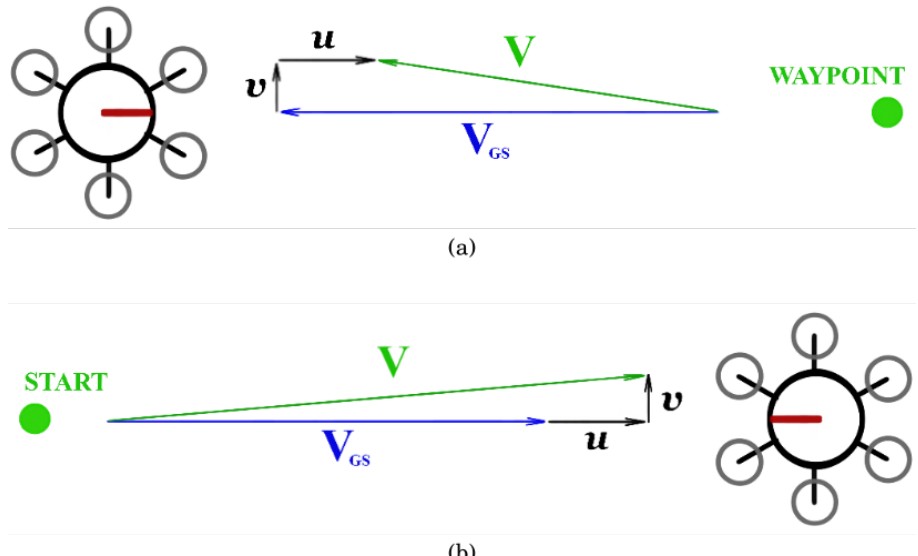

(a)

(b)

**Figure 7.** Effect of a not negligible horizontal wind $(u,v)$ during the calibration flights. **a)**: the multi-copter flies towards the waypoint: the wind disturbance adds up to the ground speed $(V_{GS})$ resulting in a lower true air speed $(V)$. **b)**: the multi-copter flies back to the starting position: in this case the wind disturbance results in a higher true air speed.

## 3.2 Data Correction

Suppose two unknown wind components $u$ and $v$ are assumed to be present at the moment of the calibration flights. In that case, the situation will be like the one represented in Fig. 7. For each imposed GS along the same direction, we can assume the wind to be constant, being the flight time relatively short. Then, at this point, Eq. (23) is used to model the UAS behaviour in the forward (F) and backward (B) sections by making use of the true air speed $(V)$. The two different tilt angles are described by:

$$\Gamma_{\mathrm{F}} = \arctan\left[K \cdot C_{\mathrm{A,F}} \cdot V_{\mathrm{F}}^2\right] \tag{26}$$

$$\Gamma_{\mathrm{B}} = \arctan\left[K \cdot C_{\mathrm{A,B}} \cdot V_{\mathrm{B}}^2\right] \tag{27}$$

$\Gamma_{\mathrm{F}}$ and $\Gamma_{\mathrm{B}}$ are the tilt angles recorded by the autopilot during the forward and backward calibration flights. The TAS for the forward and backward part are modelled by simple trigonometric relations as in Fig. 7:

$$\Gamma_{\mathrm{F}} = \arctan\left[K \cdot C_{\mathrm{A,F}} \cdot \left[(\mathrm{GS} - u)^2 + v^2\right]\right] \tag{28}$$

$$\Gamma_{\mathrm{B}} = \arctan\left[K \cdot C_{\mathrm{A,B}} \cdot \left[(\mathrm{GS} + u)^2 + v^2\right]\right] \tag{29}$$

Note that $C_{\mathrm{A,F}}$ and $C_{\mathrm{A,B}}$ are not the same values shown in Fig. 6b since those values were computed using Eq. (24). Trying to use $C_{\mathrm{A,F}}|_{\mathrm{GS}}$ and $C_{\mathrm{A,B}}|_{\mathrm{GS}}$ to solve Eq. (28) and 29 would lead to the trivial solution $(u,v) = (0,0)$.



**Table 3.** Comparison between the ERA5 reanalysis surface wind speed and the wind speed obtained by solving equations 28 and 29 (UAS WS: $\sqrt{u^2 + v^2}$), for the four calibration flights. The UAS WS is the average of the atmospheric wind speeds obtained for each GS during one mission.

| No. | Start [UTC] | ERA5 WS [ms⁻¹] | UAS WS [ms⁻¹] |
|---|---|---|---|
| **1** | 14:03 | $\leq 1$ | 0.47 |
| **2** | 14:42 | $\leq 1$ | 0.15 |
| **3** | 15:33 | $\approx 2$ | 1.60 |
| **4** | 16:06 | $\approx 2$ | 1.76 |

Thus the system (equations 28 and 29) has four unknowns ($C_{A,F}$, $C_{A,B}$, $u$, $v$) and can not be solved as it is.

In order to calculate the corrected extended drag coefficient $C_A|_{GS}$ data showed in Fig. 6b and solve the system, an average between the forward and backward data clouds is performed obtaining Fig. 8. In the same way, by using these new extended drag coefficient values, updated values for the tilt angle ($\overline{\Gamma}$) are computed by using Eq. (24). These new values represent the extended drag coefficient and the tilt angle that would have been recorded if no atmospheric wind was present during the calibration flights, namely if TAS = GS.

$$\overline{C}_A = \text{mean}\left(C_{A,F}|_{GS}, C_{A,B}|_{GS}\right) \qquad\qquad \Big|_{\forall GS} \qquad\qquad (30)$$

$$\overline{\Gamma} = \arctan\left(K \cdot \overline{C}_A \cdot V_{GS}^2\right) \qquad\qquad \Big|_{\forall GS} \qquad\qquad (31)$$

Now the new $\overline{C}_A$ values can be used to solve the system of equations 28 and 29 by approximating $C_{A,F} = C_{A,B} = \overline{C}_A$. Even though assuming equivalent values for the extended drag coefficient is still an approximation, the solution for $\Delta u$ and $\Delta v$ - now the only unknowns - is similar to the ERA5 results (Table 3 and Fig. 4).

The corrected values of tilt angle and extended drag coefficient can be now used to define our wind estimation models.

### 3.2.1   Direct model

The first model for the estimation of the horizontal wind velocity is a direct relationship between the corrected tilt angle and the tested GS. This approach has already been extensively studied (Neumann and Bartholmai, 2015) and proved to provide good results.

The fitting is performed over the updated average value of the tilt angles ($\overline{\Gamma}$) obtained from the two flight directions. The fitting function is a third-order polynomial:

$$V(\Gamma) = c_3 \cdot \Gamma^3 + c_2 \cdot \Gamma^2 + c_1 \cdot \Gamma + c_0 \qquad\qquad (32)$$

with $c_3 = -5.56 \cdot 10^{-4}$, $c_2 = 1.75 \cdot 10^{-3}$, $c_1 = 0.88$ and $c_0 = 0$ (RMSE: 0.25 ms⁻¹). The last coefficient is manually set to zero such that a null tilt angle would logically correspond to zero wind speed. The fitting function is shown by the red line in Fig.

8a.





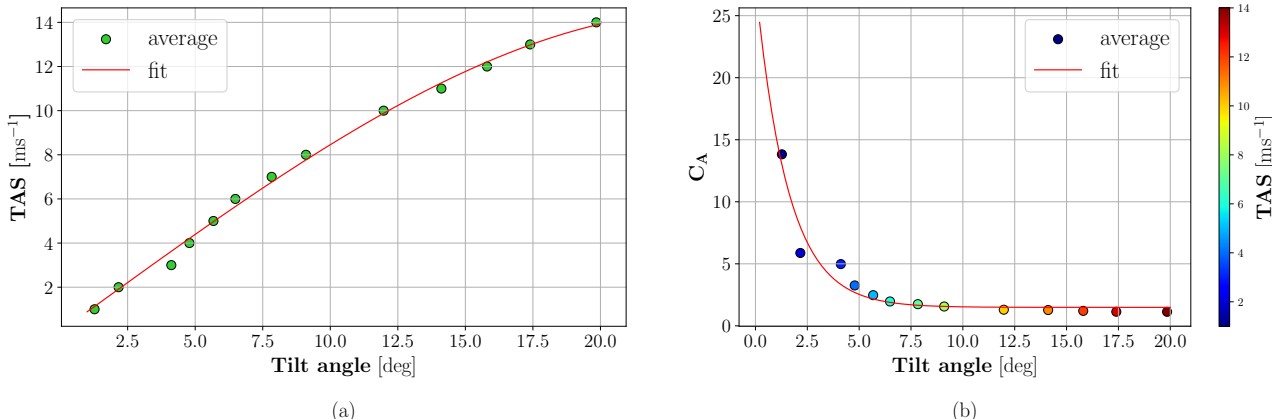

**Figure 8. a)** The green points mark the data after the post processing wind disturbance correction. The red line is the third order polynomial fit (32) that characterizes the direct model between tilt angle and horizontal wind velocity. **b)** The points mark the data for the extended drag coefficient after the post processing wind disturbance correction: the colormap represents the true air speed. The red line is the exponential decay (33) that characterizes the $C_A$ model.

This model requires only the Euler angles of the UAS. However, it is a model that represents the reality only if the same atmospheric conditions of the calibration flights are present. Any change in temperature or pressure (thus in air density), or even a change in the multi-copter mass, will offset the real horizontal wind. There is no possibility of estimating the magnitude of the offset other than performing many other calibration procedures with different payload and atmospheric conditions.

### 285   3.2.2   $C_A$ model

The second model is based on the characterization of the extended drag coefficient $C_A$ as a function of the tilt angle, similarly to the one used by Wetz et al. (2021). However, the behaviour of the average points (Fig. 8b) is better described by using an exponential decay function rather than a linear regression. The fitting is performed over the updated average value of the extended drag coefficient ($\overline{C}_A$):

$$C_A(\Gamma) = c_0 + (c_1 - c_0) \cdot e^{-\frac{\Gamma}{c_2}} \tag{33}$$

with $c_0 = 1.487$, $c_1 = 27.61$ and $c_2 = 1.55$ (RMSE: 0.81).

By using this function, it is possible to estimate the horizontal velocity by Eq. (20).

This model requires the air density and the multi-copter mass as additional inputs (see Eq. (21)). The air density is computed using pressure and temperature data, while the mass is measured using a scale before take off.

This makes the model more flexible, allowing the user to use it under different external conditions. The mass of the system is present as a stand-alone parameter in Eq. (20); however, a different payload configuration will also affect the absolute values of the calibration function $C_A(\Gamma)$. A more detailed analysis of this dependence is carried out in Sect. 5.





**Table 4.** Validation flights summary. The wind speed and direction range refers to the ultrasonic anemometer data.

| No. | Day | Start | End | Wind Speed Range | Wind Direction Range |
|---|---|---|---|---|---|
| **1** | June 17 | 9:40 UTC | 9:57 UTC | 2.0-10.0 ms⁻¹ | 90-182 deg |
| **2** | June 17 | 11:06 UTC | 11:23 UTC | 1.6-10.6 ms⁻¹ | 80-193 deg |
| **3** | June 17 | 13:11 UTC | 13:28 UTC | 0.3-8.2 ms⁻¹ | 94-187 deg |
| **4** | June 18 | 8:15 UTC | 8:32 UTC | 1.9-9.6 ms⁻¹ | 91-219 deg |
| **5** | June 18 | 9:46 UTC | 9:54 UTC | 2.6-10.1 ms⁻¹ | 103-196 deg |
| **6** | June 18 | 10:54 UTC | 11:11 UTC | 1.2-11.5 ms⁻¹ | 109-219 deg |
| **7** | June 18 | 12:56 UTC | 13:01 UTC | 3.3-12.2 ms⁻¹ | 109-175 deg |
| **8** | June 18 | 13:19 UTC | 13:36 UTC | 1.8-11.9 ms⁻¹ | 115-200 deg |

## 4 Results from validation flights

This section assesses the quality of the two models obtained with the calibration flight data. To this aim, the horizontal wind
estimation has been compared to the data provided by a Metek USA-1 ultrasonic anemometer mounted on a 99 m mast. The
comparison flights have been performed at the German Meteorological Service Boundary Layer Field Site of Falkenberg,
Brandenburg, close to the MOL-RAO observatory site, in the framework of the VALUAS project, during the FESSTVaL Field
campaign in June 2021.

The tower mounts two sonic anemometers of the same type at 50 and 90 meters altitude. These sensors provide a fast
sampling of the three wind-vector components at 20 Hz, with a measurement range from 0 to 60 ms⁻¹ and a declared accuracy
of 0.01 ms⁻¹ at 5 ms⁻¹. These sensors are considered a reliable reference since they can resolve the turbulence eddies up to half
of their sampling frequency.

Several flights have been performed over two days (17th and 18th of June 2021) of very variable atmospheric wind, covering
a range from 0.3 to 12.2 ms⁻¹. The missions consisted in hovering the UAS aside of the tower at the same altitude as the
anemometers. A safety distance of approximately 10 m was maintained between the tower and the aircraft. Eight flights were
dedicated to the wind validation (Table 4), with a total hovering time of more than one and a half hours.

The data from the sonic anemometers have been resampled to 10 Hz in order to match the multi-copter series. A cross-
correlation has been performed between UAS and tower data in order to identify an eventual time lag due to the safe distance.
The two data series have been then synchronized to delete this lag.

### 4.1 Frequency analysis

Although measuring small scale atmospheric turbulence is out of the scope of this work, a frequency analysis of the wind
estimation models has been performed. The power spectral density (PSD) of the horizontal wind is useful in order to understand
the minimum timescales of the atmospheric eddies that the multi-copter can resolve. The results displayed in Fig. 9 are the





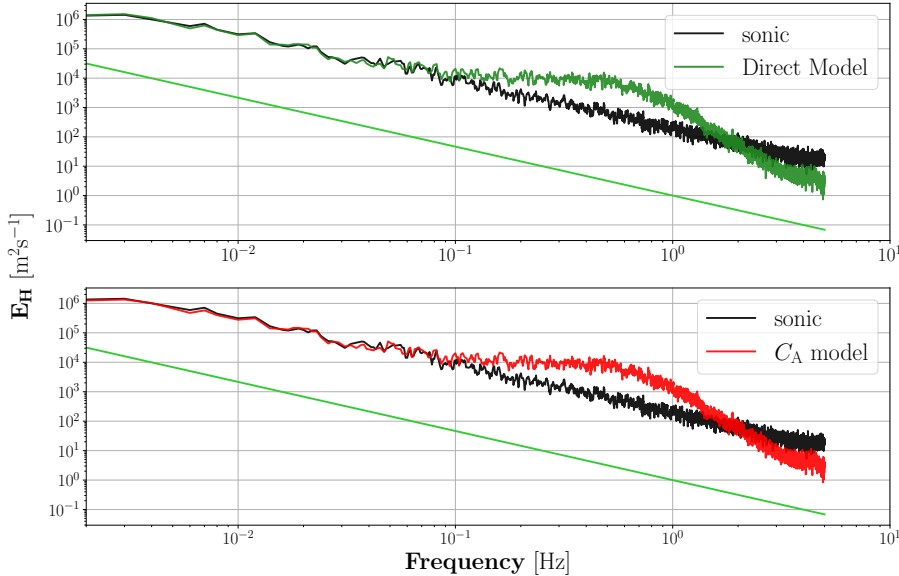

**Figure 9.** Comparison between the power spectral densities of the horizontal wind speed of the sonic anemometer (black) and the two estimation models (green and red). The light-green line is reported as a reference of the Kolmogorov -5/3 turbulence decay law.

average spectra of all our flights, computed with the raw 10 Hz UAS data. The black spectrum is computed using the sonic
anemometer data, while the green and red ones are computed using both wind estimation models. The light-green line is the reference for the Kolmogorov (1941) -5/3 decay within the inertial subrange of quasi-isotropic turbulence. Figure 9 shows an agreement between the multi-copter and the reference up to 0.1 Hz (once every 10 s). No significant differences were found for the two wind estimation methods (direct and $C_A$ model)

## 4.2   Time series analysis

Based of the outcome of the frequency analysis, data have been resampled to 0.2 Hz (once every 5 s, to avoid aliasing) in order to compare the time series: in this way, most of the undesired oscillations are removed, leading to more meaningful analysis.

The plot of the first flight at 90 m is reported in Fig. 10. During this mission the wind velocity varied between 2 and 10 $\mathrm{ms^{-1}}$. The black line in Fig. 10a represents the ultrasonic anemometer (reference), while the green and red lines are the result of the two wind estimation models. In Fig. 10b the wind direction estimation is plotted against the reference direction. The overall
agreement is satisfyingly good. A systematic discrepancy between UAS and sonics can only be seen in the wind direction (typically around 15°)

The complete wind estimation dataset for the altitude of 90 m is shown in Fig. 11. Again the reference is plotted with a black line, while the vertical black dashed lines represent the beginning and end of each flight. Flight number 1 shown in Fig. 10 is the first section of Fig. 11.



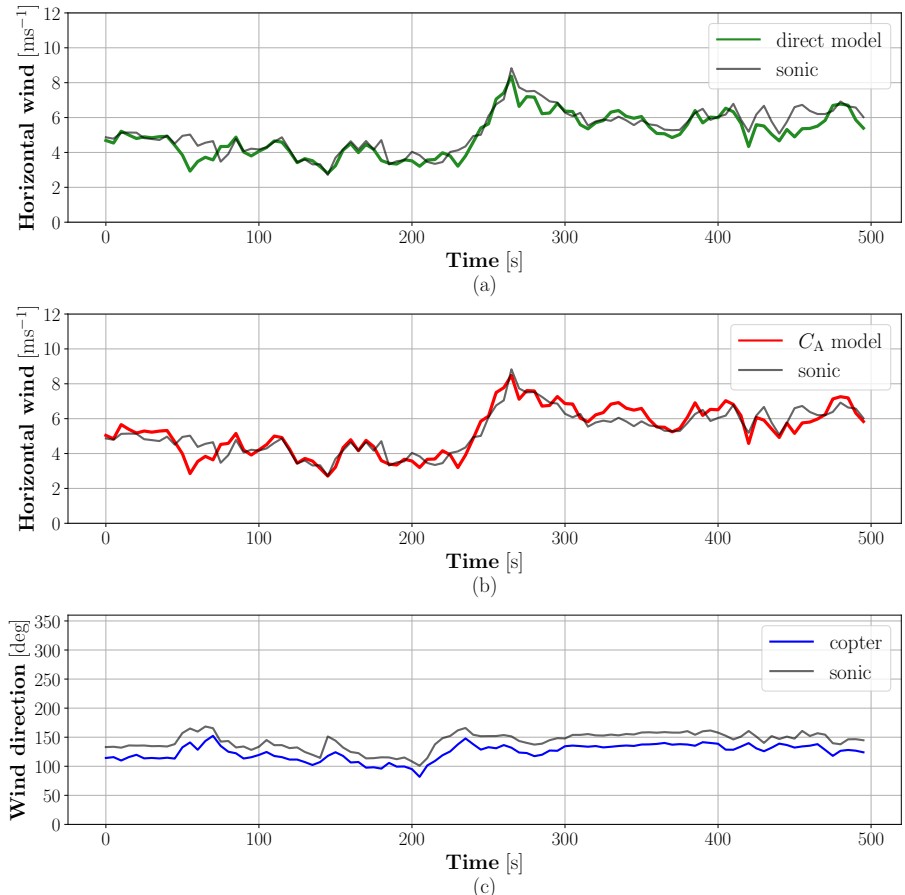

**Figure 10.** Comparison between the horizontal wind-vector detected by the ultrasonic anemometer (black) and by the multi-copter. The picture shows around 500 s of hovering at 90 m altitude for flight number 1. All the time series have been resampled to 0.2 Hz for the comparison. **a-b**): wind magnitude obtained with the two models using the UAS tilt angle (green and red). **c**): horizontal wind direction.

## 5 Discussion

### 5.1 Advantages of the multi-copter shell

The almost spherical styrofoam shell that encloses the multi-copter has several advantages. The exposed electronics are protected from precipitation and damage. But most importantly, the shell improves the wind measurement. Due to the isotropic shape, the same cross-section area is always exposed to the wind, regardless of the incoming wind direction. Neumann and Bartholmai (2015) use a quadcopter with a nearly isotropic, cylindrical fuselage for a similar approach in wind measurement and show that the radial orientation of the system to the wind direction is negligible in wind measurement. Due to the spherical shell, together with the hexacopter configuration with six rotors, which offers an even more isotropic shape compared to a

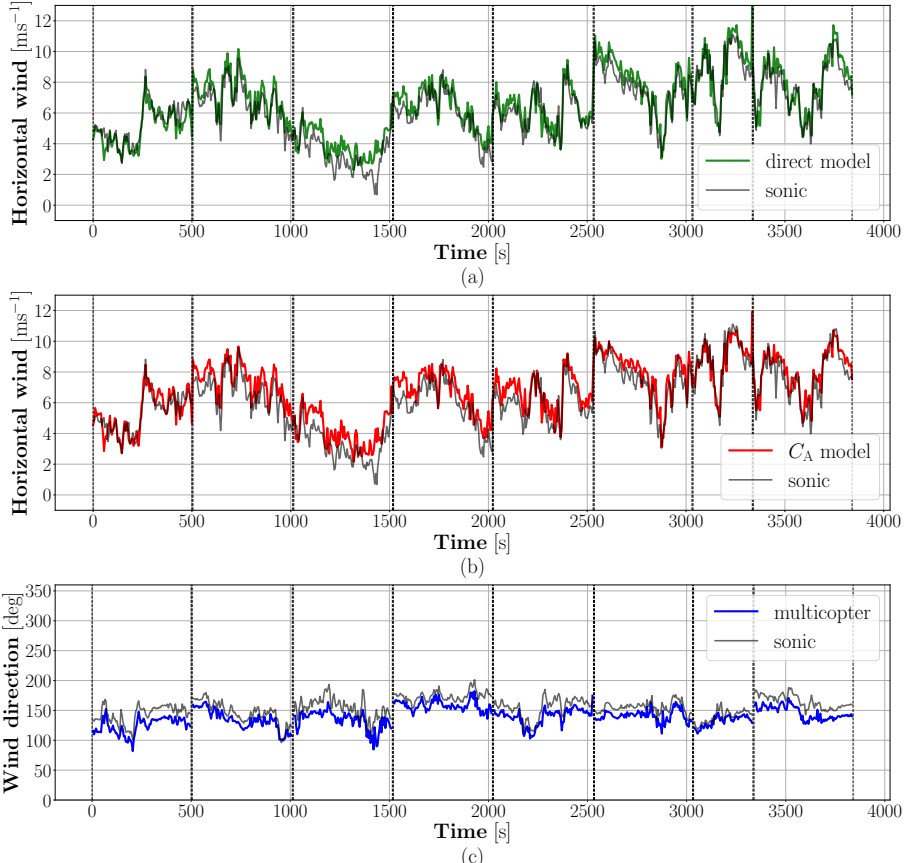

**Figure 11.** Comparison between the horizontal wind-vector detected by the ultrasonic anemometer (black) and by the multi-copter. The picture shows all the data collected at 90 m altitude (8 different flights). All the time series have been resampled to 0.2 Hz for the comparison. **a-b**): wind magnitude obtained with the two models using the UAS tilt angle (green and red). **c**): horizontal wind direction.

quadcopter with four rotors, the radial alignment to the wind can also be neglected for the system used in this work. Compared to a DJI S900 without a hull, the modified DJI S900 used here has a significantly larger fuselage cross-section and thus a higher

air resistance, with a moderate weight increase. This increases the tilt angle that the multi-copter assumes to compensate for the wind speed. Especially at low wind speeds, this means that the ratio of the signal to be measured (tilt against the flow) to unwanted signal (noise due to correction movements) is higher than with a UAS without a hull.

## 5.2 Spectra

The sonic anemometer spectrum follows the Kolmogorov -5/3 turbulence decay law up to 5 Hz, proving to be an excellent

reference for the validation. The direct and $C_{\mathrm{A}}(\Gamma)$ model's spectra are substantially similar and follow the sonic spectrum up to a frequency of $10^{-1}$ Hz. Subsequently, a sort of plateau is present up to 0.6-0.7 Hz and then eventually a steeper decay.





**Table 5.** Statistics of the wind-vector estimation. The RMSE is computed after removing the MBE between the multi-copter and ultrasonic anemometer signals.

|  | MBE | RMSE | unit |
|---|---|---|---|
| **Direct Model** | 0.32 | 0.64 | ms$^{-1}$ |
| $C_A$ **Model** | 0.57 | 0.66 | ms$^{-1}$ |
| **Direction** | -14.7 | 7.6 | deg |

Here the system acts as a low pass filter due to its physical characteristics. It means that the atmospheric wind inputs (gusts) with a frequency higher than 0.7 Hz do not manage to perturb the multi-copter for enough time to overcome its inertia.

It is important to understand why such an evident plateau is present in the central part of the multi-copter spectrum. Higher
values with respect to the sonic spectrum mean that the flying system is introducing some energy that is not due to the wind itself. The authors' opinion is that this phenomenon is a consequence of the steady and equilibrium flight hypothesis adopted while calibrating the models (Sect. 2.2.3). Indeed it was assumed that the multi-copter would respond immediately to every input in terms of horizontal wind without showing any transient.

Suppose, for example, the system to be hovering in Position Hold mode; suddenly, it receives a horizontal wind step input,
from zero to a generic value $V_{\text{step}}$. With the assumption used for building our models, the system's response should be exactly equal to the input. In reality, the system will respond when it senses an error in its position and tries to reduce it to zero. Therefore the tilt angle (and consequently the wind estimation) will undergo an overshoot with respect to the expected value for the velocity $V_{\text{step}}$ at the moment the system tries to return to its hovering position. It is possible that this dynamic introduces into the frequency range from $10^{-1}$ a 0.7 Hz some energy component not due to atmospheric turbulence.

Thus our UAS is able to resolve atmospheric eddies up to $10^{-1}$ Hz in the velocity range covered by the comparison flights. This means, taking an intermediate value of 8 ms$^{-1}$, to resolve eddies of 80 m of characteristic length. In a convective atmospheric boundary layer, e.g. in summer over land, this resolution would be sufficient to cover the most important turbulence scales.

### 5.3 Quality of the horizontal wind estimation

The mean bias error (MBE) and the root mean square error (RMSE) are presented for the two types of model in table 5. The accuracy of the two models is comparable and falls below 0.7 ms$^{-1}$ when the MBE is subtracted from the two signals. It is important to notice how the RMSE is in the same range as the previous studies even though the modified DJI S900 multi-copter is several kilograms heavier than the other systems.

The agreement with the reference signal as shown in figures 10 and 11 is good with slight differences due to the different
model formulations.

As general feedback, it is possible to note how the accuracy of both models worsens under low wind conditions. As an example in Fig. 11, in the range between 1000 and 1500 seconds we find the maximum discrepancy to the reference. As





already mentioned in the previous sections, both models are more prone to errors at the low-velocity range. The uniform band filtering of the calibration data results in a higher percentage error at low speeds. Moreover, for the $C_A$ model, the uncertainty

on the drag coefficient is higher for low calibration speeds. As a final remark, in Figures 8 it can be observed how the average of the velocity 3 ms$^{-1}$ is not as close to the fit as the others, so the fitting itself could introduce errors dependent on the wind speed.

### 5.4    Quality of the wind direction estimation

The MBE and the RMSE for the direction estimation are reported as well in table 5. The main issue in the direction estimation

is the presence of a constant offset between the multi-copter and the reference sensor of around 15 deg. The computation of the direction involved the yaw angle (Eq. (22)): it is possible that the magnetometer suffered from a deviation from its original calibration or simply that it could have been slightly moved with respect to its original position, causing the constant offset.

The multi-copter captures the wind direction variation well, and the corrected RMSE shows values lower than 8 deg.

### 5.5    Parameters' influence

#### 5.5.1    Vertical velocity and mass

These two parameters are substantially different. However, their influence on the multi-copter is, to the first level of approximation, equivalent.

The vertical wind component has been neglected so far. Nevertheless, its presence would generate a drag force directed vertically, adding or subtracting to the system's weight. Likewise, an increase or decrease of the UAS mass, for example due

to a different battery configuration or additional sensors, results in a modification of the vertical force component.

At calibration condition (denoted with subscript c), with constant flight speed/incoming wind speed, for small values of the tilt angle the following approximation is valid:

$$\Gamma_c = \arctan\left(\frac{D}{2m_c g}\right) \approx \frac{D}{2m_c g} \tag{34}$$

Then a variation in the vertical force component $\Delta F_z$ result in a modified tilt angle $\Gamma_\Delta$:

$$\Gamma_\Delta \approx \frac{D}{2m_c g + \Delta F_z} \tag{35}$$

This equation holds if the drag - the extended drag coefficient $C_A$ - does not vary sensibly with the tilt: this happens, in our case, for wind speeds higher than 8 ms$^{-1}$.

#### 5.5.2    Vertical wind speed

In the case of a vertical wind disturbance the term $\Delta F_z$ is, in usually a varying, function of time:

$$\Delta F_z = f(t) \tag{36}$$





It could be a constant term, as in the case of complex terrain (Zum Berge et al., 2021), or it could also be time-dependent, for example, in case a thermal is met. In any case, it is clear how any vertical wind component would distort the horizontal wind estimation by adding a bias: the identification of this bias is even more complicated since it may not constant throughout a single flight. The calibration flights have been performed over flat terrain. Therefore, it is unlikely to have a constant offset in the calibration data. Moreover, the amount of data gathered for each flight velocity is enough to filter out any possible vertical gust.

From a first approximation the ratio between the tilt angle at calibration conditions and the distorted tilt angle will be:

$$\frac{\Gamma_c}{\Gamma_\Delta} = \frac{m_c g + \Delta F_z}{m_c g} \tag{37}$$

Then the spoiled tilt angle will be:

$$\Gamma_\Delta = \frac{m_c g}{m_c g + \Delta F_z} \cdot \Gamma_c \tag{38}$$

Considering that the weight force of the multi-copter is around 70 N and assuming the same $C_A(\Gamma)$ relation for the vertical extended drag coefficient, a $w$ component of 5-6 ms$^{-1}$ would be needed in order to have a 10% variation in the tilt angle. Such values for the vertical atmospheric wind are infrequent. However, if the weight of the multi-copter is lower, this influence would become more pronounced.

### 5.5.3 Mass

In the case of different payload configuration, the term $\Delta F_z$ is, for electrical propulsion, a constant with respect to time:

$$\Delta F_z = \Delta m g \tag{39}$$

Similarly to the previous case, the distorted tilt angle will become:

$$\Gamma_\Delta = \frac{m_c g}{m_c g + \Delta m g} \cdot \Gamma_c \tag{40}$$

In this situation, however, 700 g are enough to achieve a 10% variation of the tilt angle. Mass variations in the order of 1 kg are totally reasonable in our specific case (a single 12.000 mAh battery weights 1.5 kg).

A variation of the multi-copter mass introduces a constant bias in the direct model (Sect. 3.2.1) since it is specific for the mass that the system had during the calibration flights. At least for high wind speeds, an increase of the mass should not affect the extended drag coefficient model as the $C_A(\Gamma)$ function becomes approximately constant. With a constant extended drag coefficient, according to Eq. (20), the parameters defining the wind speed (air density, multi-copter mass, tilt angle, and $C_A$ itself) are all independent. It is sufficient to weigh the system and insert the correct mass value in the equation. However, at low wind speeds also the $C_A$ model depends on the multi-copter mass. However, the influence is difficult to estimate since Eq. (35) becomes non-linear, and a change of the denominator will change the value of the drag force sensibly as well.





### 5.5.4 Air density

A variation of the air density modifies the horizontal component of the force system by changing the value of the drag. An estimation of the effect of this variation is not straight forward since the $C_A$ varies with the tilt angle as well.

$$\Gamma_c = \arctan\left(\frac{\rho_c V^2 C_A(\Gamma)}{2mg}\right) \approx \frac{\rho_c V^2 C_A(\Gamma)}{2mg} \tag{41}$$

If a density variation is present then:

$$\frac{\Gamma_c}{\Gamma_\Delta} = \frac{\rho_c C_A(\Gamma_c)}{\rho_\Delta C_A(\Gamma_\Delta)} \tag{42}$$

Under the conditions in which the ratio between the two drag coefficient is almost 1, so for wind speeds above 8 ms$^{-1}$ the distorted tilt angle will be:

$$\Gamma_\Delta = \frac{\rho_\Delta}{\rho_c} \cdot \Gamma_c \tag{43}$$

The calibration took place at $\rho_c = 1.181$ kg m$^{-3}$. If the system is used under different atmospheric conditions, there will be always a constant offset in the wind estimation with the direct model. For instance, in an environment with $\rho_\Delta = 0.95$ kg m$^{-3}$,
the tilt angle is reduced approximately by 20%.

This offset is avoided by using the $C_A$ model, where the air density is an input. In order to obtain this information, it is necessary to operate the UAS with some additional sensors since the autopilot alone is not designed for such measurement.

## 6 Conclusions and outlook

Two models for the horizontal wind estimation based on the tilt angle of a multi-copter while hovering are obtained by per-
forming a series of perpendicular flights at constant ground speed. By mapping several ground speeds, valuable data can be collected without using wind tunnels or meteorological masts. Particular weather conditions are required during the calibration flights: ideally, a zero wind condition should be met to increase the data quality. However, data could be used to calibrate the models even if wind is present using a post-processing method to filter out the wind disturbance components.

Increased isotropy set up, obtained by encasing the multi-copter body and electronics in a styrofoam dome, allows calculating
the tilt angle using only the roll and pitch angles. This sphere shape uniforms the aerodynamic forces with respect to the incoming wind's direction and grants a more regular change of the area exposed to the wind while the system is tilting.

Two different approaches have been used in order to generate two horizontal wind estimation models from the UAS tilt angle:

- A direct approach where the data from the calibration flights have been used in order to generate a relation between the
tilt angle and the horizontal wind speed.

- An indirect approach where the same data has been used to generate a relation between the tilt angle and the extended drag coefficient $C_A$.





Both models have been tested by comparing the wind estimation with an ultrasonic anemometer reference sensor. A frequency analysis showed that our multi-copter is able to resolve the wind speed up to a frequency $10^{-1}$ Hz, following the -5/3

Kolmogorov (1941) turbulence decay. The models showed an RMSE lower than 0.7 $\mathrm{ms}^{-1}$ for a range of velocities from 0.3 to 12 $\mathrm{ms}^{-1}$.

The direct model will always be affected by a constant offset if parameters such as the multi-copter mass or the air density change. The $C_A$ model, on the other hand, requires those parameters as input. Therefore, it is not affected by errors from air density change and is less affected by payload variations at higher speeds.

The wind direction was also computed from a direct relationship involving all the three Euler angles. The uncertainty in the yaw value seems to introduce a constant offset in the estimation; however, once the offset is identified, the RMSE falls below 8 deg.

A possible future development could be the implementation of a dual GPS module configuration. In such a way, it is possible to compute the vehicle's heading without the compass, which seems to be the primary source of uncertainty for the yaw angle.

A further development for the $C_A$ model could be achieved by performing other calibration flights with progressively increasing payload. The extended drag coefficient would then become a function of two parameters (tilt angle and mass) and the bias at lower speeds would not be an issue anymore.

Especially for lighter copters, it could be possible due to the lower inertia to sample wind at higher frequency and also correlate the power delivered from the motors to the tilt angle in order to study the vertical wind component $w$. If an increase

of the tilt angle corresponds to a decrease of the power output, it means that some wind is pushing the UAS upwards.

*Data availability.* The data are available from the author upon request

*Author contributions.* M.B. and M.S. conceived of the presented idea, designed and performed the experiments. M.B. developed the theory and performed the computations. D.S. and V.S. were involved in carrying out the experiments. M.B. wrote the paper, with contributions from M.S. and D.S. All authors discussed the results and contributed to the final manuscript.

*Competing interests.* The authors declare that they have no conflict of interest.

*Disclaimer.* This work is partly funded by the European Union Horizon 2020 research and innovation program under grant agreement no. 861291 as part of the Train2Wind Marie Sklodowska-Curie Innovation Training Network(https://www.train2wind.eu/).





*Acknowledgements.* We thank Frank Beyrich and the German Meteorological Service (DWD) for providing the infrastructure at the MOL-RAO and the ultrasonic anemometer data. The measurements in Falkenberg, which provided the data for the validation, were performed as a supplement to a Lidar validation flight project funded by the German Meteorological Service (DWD) under the funding code 4819EMF01 (VALUAS).






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
