# Peer review of "A stand-alone calibration approach for attitude-based multi-copter wind measurement systems"

_Atmospheric Measurement Techniques, 2022_

## Referee Comment (RC1)

**AMT-2022-113 / Referee Comments**

**General comments:**

The presented paper is in general well written and has a good structure. The authors have shown an interesting hardware concept to estimate wind using a UAS together with simple mathematical models and algorithms. The authors made good contributions in the analysis of each factor that affects the calculations through stand-alone calibrations and validations against conventional meteorological towers, as well as identifying and overcoming some of the short-comings. Having said that, there are some parts of the paper that deserves more attention in some parts and provide more solid concluding remarks in the Spectra analysis section. The paper can be accepted with minor revisions that must be addressed beforehand and shown below.

1. Does the paper address relevant scientific questions within the scope of AMT?

   Yes.

2. Does the paper present novel concepts, ideas, tools, or data?

   Very thorough study of an idealized case and good contributions in the mathematical analysis of factors.

3. Are substantial conclusions reached?

   Yes, although some parts need more insight and consideration of important factors inherent to the UAS.

4. Are the scientific methods and assumptions valid and clearly outlined?

   Yes. Some assumptions must be declared as belonging to the author's opinion unless there are citations about it.

5. Are the results sufficient to support the interpretations and conclusions?

   Yes.

6. Is the description of experiments and calculations sufficiently complete and precise to allow their reproduction by fellow scientists (traceability of results)?

   Yes.

7. Do the authors give proper credit to related work and clearly indicate their own new/original contribution?

   Yes.

8. Does the title clearly reflect the contents of the paper?

   Yes.

9. Does the abstract provide a concise and complete summary?

   Yes.

10. Is the overall presentation well structured and clear?

    Yes.

11. Is the language fluent and precise?

    Yes. Some minor corrections though.

12. Are mathematical formulae, symbols, abbreviations, and units correctly defined and used?

    In general yes. Equations 10-12 deserves attention as described below.

13. Should any parts of the paper (text, formulae, figures, tables) be clarified, reduced, combined, or eliminated?

    See comments below.

14. Are the number and quality of references appropriate?

    Yes.

15. Is the amount and quality of supplementary material appropriate?

    N/A

**Other comments:**

**Line 19-23:** There are a few papers that compares conventional meteorological instruments with the UAS-based weather instruments. Please add some references and elaborate a bit more (1 or 2 sentences). Here are some papers as examples (but not limited to):

Pinto, J. O. et al. "The Status and Future of Small Uncrewed Aircraft Systems (UAS) in Operational Meteorology" Bulletin of the American Meteorological Society, 102(11), E2121-2136, https://doi.org/10.1175/BAMS-D-20-0138.1, 2021.

Bell, T. M. et al. "Confronting the boundary layer data gap: evaluating new and existing methodologies of probing the lower atmosphere", Atmos. Meas. Tech., 13, 3855–3872, https://doi.org/10.5194/amt-13-3855-2020, 2020.

**Line 31:** Edit to say: "The effects of incoming wind on the UAS attitude provides the basis for …". The word dependency sort of contradicts the statement of "fly autonomously without prior knowledge of the wind field" in line 24.

**Line 39:** Instead of feasible, say something like "more affordable/accessible/simple". Wind tunnels are feasible too, but the presented method requires less "effort" to achieve the same goal.

**Line 40:** Add the following: "… repeated for different multi-copters under the same conditions".

**Table 1:** Make first letter upper case to be consistent with the other tables in the document.

**Line 67:** Please provide the version of ArduCopter used.

**Line 70:** Since the styrofoam is crutial component in this study, it would be worth describing it more in depth. Please provide with some more properties about the material and a brief description of the construction of the enclosure.

**Line 73-75:** Table 1 is not referenced anywhere in the text. You may want to mention it here.

**Line 105:** However, yaw is important for estimating the wind direction. I'm assuming that the reason of doing yaw = 0 here is just for the computation of wind speed which is a function of the tilt only. Please clarify this if so.

**Line 114:** replace used with applied

**Equations 10-12:** I understand what you did here. However, many readers will be confused by the appearance of the wind components u and v. To avoid confusion, you can define **V** = **V**wind - **V**uas = **V**eq = {u,v,w} knowing that **V**uas = 0 because of hovering conditions.

**Line 152-154:** I think this statement has to be declared as an author's opinion and the approach taken by them, please do so. In my opinion, I believe A(Tilt) can resolve forces of the solid body alone (styrofoam shpere), while the external components (mainly the rotors) need other different approaches, like a different model to describe them, and then combine the results. Please refer to the following paper to understand other points of view in the subject:

Wang, J.Y. et al. "A wind estimation method with an unmanned rotorcraft for environmental monitoring tasks" Sensors, 18(12), 4504, https://doi.org/10.3390/s18124504, 2018.

Rajan, G and D'Andrea, R "Computationally Efficient Force and Moment Models for Propellers in UAV Forward Flight Applications" Drones, 3(4), 77, https://doi.org/10.3390/drones3040077, 2019.

**Line 165:** Replace "will maintain its position as computed by" with "holds its position assisted by"

**Line 174-178:** For non-zero wind conditions, an idea that you can try in the future is to let the drone drift by the wind for a period of time. Eventually, the drone velocity will match the wind speed (in other words, wind = ground speed) and you can use that info in a slightly variation of your calculations to obtain the drag coefficients.

**Line 188-189:** Please also mention authorizations and permissions to fly drones in the area, if any. It is good example to show the reader the efforts made to fly drones legally.

**Line 195:** Replace "situation" with "conditions".

**Line 236:** Replace "A" with "The", and "end up in obtaining" with "produce".

**Line 236-240:** In theory, these "sloped lines" should ideally be just points. You can prove this in simulations within ideal environments. However, your data looks spread out because the drone is tilting back and forth correcting its GS and even flighting turbulence in its way (like its own propeller wash). The lower the GS, the more spread it looks because speed is getting close to the minimum velocity resolution of the GNSS and that introduces errors. The authors continue explaining this in lines 260-264, but without much depth. It would be good to have a physical meaning to all this. Please consider adding it (2-3 sentences more should be enough).

**Section 3.2:** Have you considered doing circular flights? Since the drone is a sphere and you are assuming isotopic conditions, a circular flight fashion should be valid. This will also help removing errors caused by wind from the calculations by taking the tilt average around the circle.

**Line 283:** Replace "will offset the real horizontal wind" with "will produce an offset in the horizontal wind estimates".

**Line 287-288:** Again, this statement has to be declared as an author's opinion and the approach taken by them, please do so. Unless you found literature that also supports this, please cite them.

**Lines 292:** Replace "by" with "described in".

**Line 293:** I'm assuming that you are computing the density of dry air since you only mention pressure and temperature variables. If you have humidity data, it is possible to also include the water vapor density for improved accuracy.

**Section 5.2:** The discussion and analysis of the spectra results are quite consistent and valid. However, there are other factors that are relevant and deserves some discussion too. The propeller wash can very well be within the frequency range of the plateau, injecting energy into the surrounding air. Its effects should be greater when the wind is low since the prop wash doesn't move away. But this is hard to see in Figure 9 because it shows an "average" of all the flights and wind velocities. GNSS is also another factor, if the position estimation drifts away, the drone will try to follow the wrong position estimate and tilt towards it. The drone is basically rocking back and forth, and this is reflected as increased energy in that range of the spectra.

It is interesting to note that the energy then decreases to levels close to the sonic anemometer spectra, to me this means that the drone's high frequency wind estimates hasn't reached the noise floor yet and be as sensitive as the sonic anemometer. Unless there is some kind of artifact in the algorithm. I suggest the authors to explore more on this and come back with conclusions.

**Section 5.4:** Did you check if the compass measurements are with respect to true north? However, 15deg seems high. ArduPilot offers advance magnetometer calibrations and it compensates for the induced magnetic fields of the electronics. Consider this for the next experiments.

**Line 404:** replace "usually" with "general". The word "varying" can be omitted.

**Line 408:** Correct sentence "since it may not be constant"

**Line 414:** "Undesired" instead of "spoiled" would sound better.

**Line 427:** Consider replacing "since it is specific for the mass that the system had" with "since the mass is unchanged during the calibration flights"

**Line 455-456:** Rephrase "uniforms the aerodynamic forces with respect to the incoming wind's direction" to "helps producing uniform aerodynamic forces with respect to the incoming wind from any direction"

---

## Referee Comment (RC3)

The manuscript presented by Matteo Bramati et al. proposes a stand-alone calibration technique for estimating wind velocity profiles in the lower atmosphere. This topic is of high relevance to the atmospheric science and engineering communities as new advancements in UAS capabilities can help improve the spatiotemporal resolution of wind velocity observations that are critical for characterizing the evolution of the atmospheric boundary layer. However, the manuscript does not advance significantly the state of the art of wind estimation in its current form. Therefore, I cannot recommend this manuscript for publication in the AMT journal.

**Does the paper present novel concepts, ideas, tools, or data?**

The wind estimation concept presented in this manuscript lacks novelty. Already, previous studies have explored the use of point mass models to infer the horizontal components of wind velocity. Moreover, what is presented as a stand-alone calibration process to characterize tilt as a function of air-relative velocity have already been performed Palomaki et al., as well as Gonzalez-Rocha et al.

Palomaki, R.T., Rose, N.T., van den Bossche, M., Sherman, T.J. and De Wekker, S.F., 2017. Wind estimation in the lower atmosphere using multirotor aircraft. *Journal of Atmospheric and Oceanic Technology*, *34*(5), pp.1183-1191.

Donnell, G.W., Feight, J.A., Lannan, N. and Jacob, J.D., 2018. Wind characterization using onboard IMU of sUAS. In *2018 Atmospheric Flight Mechanics Conference* (p. 2986).

González-Rocha, J., Woolsey, C.A., Sultan, C. and De Wekker, S.F., 2019. Sensing wind from quadrotor motion. Journal of Guidance, Control, and Dynamics, 42(4), pp.836-852.

Abichandani, P., Lobo, D., Ford, G., Bucci, D. and Kam, M., 2020. Wind measurement and simulation techniques in multi-rotor small unmanned aerial vehicles. IEEE Access, 8, pp.54910-54927.

**Are substantial conclusions reached?**

The author's claim to present a technique that does not require the use of a wind tunnel or mast towers. However, the validation experiments discussed in Section 4 were performed using a sonic anemometer, a standard practice for validation sUAS wind estimates (see references below).

Nolan, P.J., Pinto, J., González-Rocha, J., Jensen, A., Vezzi, C.N., Bailey, S.C., De Boer, G., Diehl, C., Laurence, R., Powers, C.W. and Foroutan, H., 2018. Coordinated unmanned aircraft system (UAS) and ground-based weather measurements to predict Lagrangian coherent structures (LCSs). *Sensors*, *18*(12), p.4448.

Barbieri, L., Kral, S.T., Bailey, S.C., Frazier, A.E., Jacob, J.D., Reuder, J., Brus, D., Chilson, P.B., Crick, C., Detweiler, C. and Doddi, A., 2019. Intercomparison of small unmanned aircraft system (sUAS) measurements for atmospheric science during the LAPSE-RATE campaign. Sensors, 19(9), p.2179.

**Are the scientific methods and assumptions valid and clearly outlined?**

In addition to developing a model-based wind estimation technique, the authors propose simplifying the aerodynamic characteristics of sUAS by enclosing the airframe and electronic components using a Styrofoam sphere. The authors implicitly assume the airframe drag effects to be significant. However, data that support this assumption are have not been presented. On the other hand, a previous study by Powers et al. has shown multirotor sUAS drag effects to be dominated by the propeller and airflow interactions instead of airframe shape. Moreover, quadrotor experiments performed by González-Rocha et al. show the tilt variations as a function to sideslip angle to be within the noise of the measurement at different ground speeds.

Powers, C., Mellinger, D., Kushleyev, A., Kothmann, B. and Kumar, V., 2013. Influence of aerodynamics and proximity effects in quadrotor flight. In *Experimental robotics* (pp. 289-302). Springer, Heidelberg.

González-Rocha, J., Woolsey, C.A., Sultan, C. and De Wekker, S.F., 2019. Sensing wind from quadrotor motion. Journal of Guidance, Control, and Dynamics, 42(4), pp.836-852.

**Are the results sufficient to support the interpretations and conclusions?**

The authors need to perform experiments to compare the inflow angle of nominal and spherical sUAS configurations over a range of ground speeds and sideslip angles.

**Is the description of experiments and calculations sufficiently complete and precise to allow their reproduction by fellow scientists (traceability of results)?**

Yes, the description of experiments and calculations are in general complete. However, there are formulae that need to be improved for correctness.

**Do the authors give proper credit to related work and clearly indicate their own new/original contribution?**

Tilt models to estimate wind velocity have been proposed before. It was difficult to understand how the work presented in this manuscript improves upon previous models.

**Does the title clearly reflect the contents of the paper?**

No, the wind estimation algorithm being presented is not a stand-alone technique. The implementation of this algorithm requires calibration experiments next to a conventional wind sensor.

**Does the abstract provide a concise and complete summary?**

The abstract does not provide a concise and complete summary of the work presented. It was difficult to appreciate what the authors

**Is the overall presentation well structured and clear?**

The presentation of the manuscript is well structured. However, there are sections of the manuscript that need to be improved for clarity and conciseness.

**Is the language fluent and precise?**

The authors can significantly improve the language to be more precise.

**Are mathematical formulae, symbols, abbreviations, and units correctly defined and used?**

The formulae need to be revised. For example, in Eqs. (1) and (2) the rotation matrices need to be defined. Additionally, the transformation presented in Eq (1) need to be transposed for correctness. Moreover, the tilt angle in Eq (3) can be estimated using the product rule.

**Should any parts of the paper (text, formulae, figures, tables) be clarified, reduced, combined, or eliminated?**

The abstract language needs to be clarified. As it stands, it is not evident that the authors are proposing a method based on flight transects for characterizing a wind estimation tilt model instead of hovering inside of a wind tunnel or next to a sonic anemometer.

**Are the number and quality of references appropriate?**

The manuscript does not present a comprehensive survey of model-based estimation techniques.

**Is the amount and quality of supplementary material appropriate?**

Yes.

---

## Author Comment (AC1)

**AR1 Reply**

**Line 19-23:** *There are few papers that compares conventional meteorological instruments with the UAS-based weather instruments. Please add some references …*

Thanks for the suggestions. We added some sentences in the introduction, and found another paper comparing UAS with radiosondes.

**Line 31:** *Edit to say: "The effect of the incoming wind on the UAS attitude provides the basis for …" …*

Done.

**Line 39:** *Instead of feasible say something like "more affordable/accessible/simple". …*

Changed with more accessible.

**Line 40:** *Add the following: "… repeated for different multi-copters under the same conditions".*

Done.

**Table 1:** *Make first letter upper case …*

Done.

**Line 67:** *Please provide the version of ArduCopter used.*

Done.

**Line 70:** *Since the styrofoam is crucial component in this study, it would be worth describing it more in depth. …*

Added a short description of the sizes of the dome and the way we secure it on the copter body.

**Line 73-75:** *Table 1 is not referenced …*

Done at the beginning of section 2.1.

**Line 105:** *However, yaw is important for estimating the wind direction. …*

We added a sentence to clarify that the yaw is ignored at this point only because we are looking for the tilt angle. It will be used later for the direction.

**Line 114:** *Replace used with applied.*

Done.

**Equations 10-12:** *I understand what you did here. However, many readers will be confused by the appearance of the wind components u and v. To avoid confusion you can define $V = V_{wind} – V_{uas} = V_{eq} = \{u,v,w\}$ knowing that $V_{uas} = 0$ because of hovering conditions.*

We agree that the u and v notation is not clear due to the fact that later on they are addressed as the wind speed components. However, equations 6 and 7 from Gonzalez-Rocha et al.2017 describe the behavior of the UAS flying in still air. If atmospheric wind is present some other terms appear because of the composition of the two velocity vectors. We decided to start from this set of equations because they are simpler and it is quite intuitive to switch from a no-wind condition flight and hovering.

We have reformulated the equation in order to remove the u and v variables by using the notation x_dot for u and x_dotdot for the acceleration in x direction. We adapted accordingly the following assumptions and definitions.

**Line 152-154:** *I think this statement has to be declared as an author's opinion and the approach taken by them, please do so. In my opinion, I believe A(Tilt) can resolve forces on the solid body alone (styrofoam sphere) while the external components (mainly the rotors) need other different approaches, like a different model to describe them, and then combine the results.*

The variable Ca(Tilt) accounts for everything we can not assume to be constant at different velocities. Our shape being a sphere, A(Tilt) would be a constant value if it would only resolve the forces on the sphere. We do not want to model the rotors in order to keep this approach accessible for every type of copter without running a model each time.

Wang et al. Is an interesting paper. Their method seems to overcome the wind estimation methods by tilt angle only. However, they similarly measure three drag coefficients as we do (flying backward and forward) in an indoor environment. There are some problems, though: they assume these coefficients to be referred to the body axes. However, this approximation is valid only when the Tilt angle (the wind speed) is low.

When the tilt angle gets bigger, the difference between the inertial and body reference frame becomes non-negligible. Then measuring this type of drag coefficients would probably be possible only with a wind tunnel equipped with a scale and holding the copter *untilted* for multiple wind tunnel speeds.
But let's assume we could somehow measure the three drag coefficients referred to the body axes. They still might not be constant in a range of wind speeds we would operate the copter afterward. Indeed the authors show results only for a simulation and experiment with 1 ms-1 wind speed (this also allows them to assume a linear relation between aerodynamic forces and relative velocity). We think Step 11 of their algorithm could benefit from a model like the one we present for the Ca in our manuscript.

Rajan and D'Andrea developed two different models for the forces and moments generated by a single rotor in forward flights. This model in its simplest form is a function of 9 geometric parameters of the blades and three flights parameters (RPM, V, and tilt). It is already a great effort to find the 9 parameters, then the RPM of our system is not available in our dataset. Moreover the velocity that each one of the rotors of our system experience might not be the same (some of them might be in the lee of the sphere).

**Line 165:** *Replace "will maintain its position as computed by" with "holds its position assisted by".*

Done.

**Line 174-178:** *For non-zero wind conditions, an idea that you can try in the future is to let the drone drift by the wind for a period of time …*

We have read some papers that perform this method. Still, as far as we see, there are two significant problems: we can not control the UAS path and can not systematically test a lot of ground speeds. Thus the calibration would be limited to the wind speeds present during that specific daty. However, leaving the multicopter in altitude mode is effectively the only way one could get the body axes' drag coefficients without using a wind tunnel (Wang et al. Coefficients).

**Line 188-189:** *Please also mention authorizations and permissions to fly drones in the area, if any. It is good example to show the reader the efforts made to fly drones legally.*

Added a couple of sentences about the permission we have from the airfield itself.

**Line 195:** *Replace "situation" with "conditions".*

Done.

**Line 236:** *Replace "A" with "The", and "end up in obtaining" with "produce".*

Done.

**Line 236-240:** *In theory, these "sloped lines" should ideally be just points. You can prove this in simulations within ideal environments. However, your data looks spread out because the drone is tilting back and forth correcting its GS and even fighting turbulence in its way (like its own propeller wash. The lower the GS, the more spread it looks because speed is getting close to the minimum velocity resolution of the GNSS and that introduces errors. …*

The GS is maintained with a good accuracy by the autopilot as it can be seen in Figure 5. Here the idea is just that by definition every tested velocity defines a line in Figure 6b and the slope of the line depends on the GS. The lower the GS the steeper the line. So with comparable tilt variance we

get a very high Ca variance at low speeds while a limited variance at high speeds. It is difficult to see a physical meaning.

**Section 3.2:** *Have you considered doing circular flights? …*

This is a really nice suggestion. Performing circular flights is possible with ArduPilot and also one could control where the copter should face during these flights (in order to see if any difference is present in terms of incoming wind direction). The only problem is that the force system modeling is a bit more involved since also centripetal force has to be considered and an eventual atmospheric wind also has to be taken into account. We have already performed some of these flights with a different system and we will have a look at the data soon. Probably there will be  a dedicated section about this topic in a future manuscript of a colleague of mine.

**Line 283:** *Replace "will offset the horizontal wind" with "will produce an offset in the horizontal wind estimates".*

Done.

**Line 287-288:** *Again this statement has to be declared as an author's opinion and the approach taken by them, please do so. Unless you found literature that also supports this, please cite them.*

Modified the sentence. We did not find any source in the literature, the idea is that by looking at the data they clearly do not show a linear behavior, so a non linear function was preferred.

**Line 292:** *Replace "by" with "described in".*

Done.

**Line 293:**  *I am assuming that you are computing the density of dry air since you only mention pressure and temperature variables. If you have humidity data, it is possible to also include the water vapor density for improved accuracy.*

Yes it is dry air. For the dataset described in the paper we did not have humidity measurements yet. Now the system has an array of sensors and we get pressure temperature and humidity data from the multi-copter itself. We will use also the humidity data in the future.

**Section 5.2:** *The discussion and analysis of the spectra results are quite consistent and valid. However, there are other factors that are relevant and deserves some discussion too. The propeller wash can very well be within the frequency range of the plateau, injecting energy into the surrounding air. Its effect should be greater when the wind is low since the prop wash does not move away. But this is hard to see in Figure 9 because it shows an "average" of all the flights and wind velocities. GNSS is also another factor, if the position estimation drifts away, the drone will try to follow the wrong position estimate and tilt towards it. The drone is basically rocking back and forth, and this is reflected as increased energy in that range of spectra.*

*It is interesting to note that the energy then decreases to levels close to the sonic anemometer spectra , to me this means that the drone's high frequency wind estimates has not reached the noise floor yet and be as sensitive as the sonic anemometer. Unless there is some kind of arifact in the algorithm. I suggest the authors to explore more on this and come back with conclusions.*

We have checked the spectra for each single hovering section during the validation flights. The plateau is more or less visible in all of them. It is challenging to understand if the rotor downwash plays a role here and up to which wind speeds. Even during the same hovering mission, the wind changed quite significantly (see  Figure 10c where it changes from 4 to 9 ms-1).

While in PosHold mode, the UAS uses its EKF to maintain its position (GNSS is used as input to the EKF together with other sensors). We have checked the output of the autopilot EKF in terms of position. Over the whole hovering time during the calibration flights the standard deviation of the distance to waypoint is around 8 cm. From the autopilot we can not see any sign of position estimation drifts.

**Section 5.4:** *Did you check if the compass measurements are with respect to the true north? However 15 deg seems high. ArduPilot offers advance magnetometer calibrations and it compensates for induced magnetic fields of the electronics. Consider this for the next experiments.*

We did not perform a calibration with a current compensation, however the bias was in the range of other studies. We are planning for the future to use a differential GPS configuration in order to improve the wind direction estimation.

**Line 404:** *replace "usually" with "general". The word "varying" can be omitted.*

Done.

**Line 408:** *correct the sentence "since it may not be constant".*

Done.

**Line 414:** *"undesired instead of "spoiled" would sound better.*

Done.

**Line 427:** *consider replacing "since it is specific for the mass that the system had" with "since the mass is unchanged during the calibration flights".*

Done.

**Line 455-456:** *Rephrase "uniforms the aerodynamic forces with respect to the incoming wind's direction" to "helps producing uniform aerodynamic forces with respect to the incoming wind from any direction.*

Done.

---

## Author Comment (AC2)

**AR2 Reply**

**General comments:**

*- The title of the manuscript is a "stand-alone calibration approach". But why is an in-flight calibration actually necessary? What is the robustness, especially compared to dedicated calibration measurements? In general, the calibration flights that are presented are not well described. A time series of wind speeds during the calibration would be necessary to judge stationarity of the flow during calibration. What are the limits of the calibration approach in terms of wind speed and turbulence conditions? What are the expected uncertainties?*

There are different approaches to calibrate a multicopter for wind measurements. Using a wind tunnel to map the tilt of the copter at different flow speeds is an option. Still, of course, it requires a wind tunnel, which is very expensive, and sometimes it has a long waiting list for availability. Moreover, for medium size copters (like the S900 we used), one would need a vast test chamber to avoid disturbances from the wall, especially at low speeds. Another option is to hover next to a sonic anemometer out in the field. It might be a very accurate way to build a model since one already has a precise reference (the sonic). However, it is impossible to control the atmospheric wind. Therefore creating a model covering all possible wind speeds would be a very complex task.
With the method proposed in this study, it is possible to map the complete range of flight speeds at a very low cost with comparable results in terms of accuracy. (I added some lines to clarify these points in the introduction and Section 2.2.5.)

Unfortunately, a time series of the wind speed at 50 m altitude is unavailable at the moment of the calibration flights. However, we built our method so that no other instrument is needed. We can estimate the wind speed with equations 28 and 29 for each forward and backward flight. The duration of a single forward and backward path is approximately 2.5 minutes. We believe the average wind can be considered stationary over this time window, and small-scale turbulence can be filtered out by taking the mean value of the tilt angle. (See also comment below about surface wind speeds).

The limit in terms of average wind speed for the calibration flights is when the copter can not hold its GS anymore under the effect of the external wind. In our case, the S900 can fly safely up to 20 ms-1. Then our limit in terms of wind speed is 6ms-1 if we want to calibrate up to 14ms-1. This calculation does not consider possible gusts, but it is a basic example to explain the concept. For turbulence, it is difficult to identify a precise threshold. It is more a matter of flight safety and whether the autopilot can stabilize the flight under gusty conditions.

The main uncertainties of our method lie in the steady equilibrium hypothesis made in the model formulation. We assume that the copter tilt angle represents the actual wind speed at any time, but this neglects any UAS transient. (Added section 5.6).

*- How does this calibration approach differ from a calibration approach that is described for the PX4 autopilot:https://docs.px4.io/v1.9.0/en/advanced_config/tuning_the_ecl_ekf.html#mc_wind_estimation_using_drag ?*

The calibration proposed by this library does not take into account wind disturbances. Moreover, after the complete procedure, the ballistic coefficient is set to be a single value. So, it is assumed that this value does not change with the Reynolds number or tilt angle. Our dataset shows that the terms that compose the ballistic coefficient change quite significantly over the range of flight speed of the multicopter.

*- Does the sphere really create rotationally symmetric flow around the multicopter? The rotors are a very important part of the aerodynamic features of the UAS and probably have more effect than the frame shape. I would at least have expected a graph showing the calibration error versus flow direction. There is probably not enough data there for this purpose, but I do not think that any conclusive statement can be made without such tests.*

We recently performed a test in order to prove this. The result is now shown in section 5.1.

**Specific comments:**

**p.1, l. 7**: *isotropy the right word? I do not think it should be used for the shape, but for the attributes of an object / material. Example: a perfect wooden shpere is anisotropic, because its material properties like shear stresses etc are different depending on the grain direction.*
We changed isotropy with symmetry.

**p.1, l.17**: *sonic anemometers do not need wind vanes, because they provide at least the two-dimensional wind vector.*
Yes, we agree, we changed the sentence, the order of the words was wrong and caused confusion.

**p.2, l.47:** *"linear behaviour" I am not sure what is meant by a linear behaviour of the parameter and I do not believe that Wetz et al. do assume a linear behaviour of the drag equation.*
They do not assume linear behavior of the drag equation, however they model the UAS drag coefficient using a linear model: Cd A = Cd0 A0 + cp * tilt. Instead of "parameter" we will use again "coefficient" to be clearer.

**p.2, l.50**: *I would prefer "symmetry" over isotropy, see above.*
Changed.

**p.2, l.56**: *multicopter*
We changed all multi-copter to multicopter throughout the paper.

**p.7, l.165:** *PosHold mode is not explained.*
We added a short explanation similar to what mentioned in the official ArduPilot documentation.

**p.8, l.186**: *Were surface wind speeds measured? If so, where and how?*
We had an anemometer placed close to the ground station and measured for the whole time of the missions. The picture below shows the 10min average wind speed and includes the mission sequence.

[Figure]

From this picture the wind is constant during the single forward and backward leg for each GS.

**p.9, l.206f**: *"disturb the UAS with the same magnitude in an opposite way..." This assumes that wind is constant during the flight, but during an afternoon with low wind speeds, radiation input and probably turbulence, this assumption can be violated significantly.*
As shown in the picture in the previous comment, the wind at the ground is constant along the same GS forward and backward leg. We are not assuming a constant wind during the whole 20 minutes flight but only along a single forward and backward pair of legs. A forward and backward test lasts approximately 2.5 minutes, as seen in Figure 5, which allows assuming a constant wind speed during this time window.

**Fig.4:** *There seems to be no scale or information about the position and extent of the map.*
We updated the figure adding the lat lon ticks.

**p.11, l.222**: *what is 8-1 ms^{-1}?*
It is the range of velocities mapped during the first mission, we have changed the content of the parentheses to be clearer.

**p.12, l.239f**: *I do not understand this statement "around 10 for 1 ms^{-1}". 10 what?*
It refers to the Ca coefficient. The coefficient has no unit, being dimesionless.

**p.13, l.247f**: *"wind to be constant". It would be good if you showed this with independent measurements. Turbulence occurs on a wide range of scales and can significantly disturb scales within the flight time.*
See comments **p.8, l.186, p.9, l.206f**

**Table 3**: *I am much concerned about the comparison of ERA 5 data with a grid size of 9km and local measurements in quite complex, heterogeneous terrain as in Poltringen. Can it really be expected to match?*
Our intention is that the ERA 5 data gives an overview about the synoptic situation including the wind speed and direction during the measurements flights. In addition, the terrain around Poltringen is fairly flat without obstacles around it, so we assume the ERA5 wind gives a solid estimate of the wind situation in Poltringen.

**p.14, l.269**: *but is ERA5 a good comparison?*
We do not want to use the ERA5 data to compare the copter measurements directly, but more as an indicator that we had indeed a low wind situation at the beginning while higher mean wind for the last two missions (see also comment above).

**p.14, l.278 and p.15, l.291**: What does this RMSE represent?
It is the RMSE value between the fit function and the original data. We opted for RMSE instead of the more common R squared since the drag coefficient model has been fitted by using a non-linear function.

**p.16, l.306**: *every sensor can only resolve signals up to half of the sampling frequency according to the Nyquist theory.*
The sentence was unclear, the point here is that we have a reference that provides valid data up to 10Hz which is exactly the same frequency our autopilot logs data. We changed the sentence to be clearer.

**p.16, l.313**: *but the time lag depends on the wind speed. Has this been considered?*
The time lag here has been calculated for each hovering section. It represents an averaged time lag over the whole time of hovering next to the sonic. In some cases, the wind speed changes sensibly; however,

we consider the usage of this lag conservative enough in terms of data quality. In other words, if we would calculate the real time lag for each instant, we would most likely get a better result in terms of RMSE and not worse.

**Sec.4.1:** *As the text says, turbulence is out of scope of the work. I that context I wonder what the section and Fig. 9 add to the goals of the manuscript. Maybe they can be removed. If it is only used to show that there is noise above 0.2 Hz and data needs to be rejected above that frequency, this could be mentioned early in the data processing.*

These plots were extremely useful for us in the post-processing procedure, since only thanks to this analysis we could correctly chose the resampling frequency for our data. Therefore we suggest to keep this Section and Figures.

**p.20, l.356ff**: *I have doubts about these opinions and theories. I think they should not be presented without evidence, because it could confuse readers. A simple explanation for a plateau followed by a steep decay often is a noise level in the dynamics (e.g. by vibrations) and low-pass filters in the senors at higher frequencies (here the Kalman Filter of the angle solution).*

This section has been removed since we have not enough data to support our statements

**p.20, l.370:** *It is unclear how MBE and RMSE are calculated. What are the considered averaging periods. What is the cause of the MBE?*

The MBE and RMSE reported in table 5 are calculated using all the data we gathered while hovering close to the sonic. Of course, the parameters are computed on the 0.2Hz resampled data. Also, as mentioned in the text, the RMSE is calculated by subtracting the MBE between the two signals.

The cause of MBE is challenging to identify. Also, previous literature does not explain why a bias in the real environment is still present after proper calibration. Regarding the direct model, any change in multicopter payload, or air density might cause this bias. On the other hand, for the Drag coefficient model, the bias caused by the air density is not a problem anymore. In contrast, the one caused by the mass is still partially an issue. In the end, the slightly higher bias for the Ca model might result from a non-perfect fit of the Ca points by the exponential decay.

**Sect. 5.5:** *I do not understand why there is a subsection "vertical velocity and mass" and then subsections for each of those terms on the same level.*

We changed the structure of the last sections so that they are not more on the same level.

---

## Author Comment (AC3)

**AR3 Reply**

**Does the paper present novel concepts, ideas, tools, or data?**

*The wind estimation concept presented in this manuscript lacks novelty. Already, previous studies have explored the use of point mass models to infer the horizontal components of wind velocity. Moreover, what is presented as a stand-alone calibration process to characterize tilt as a function of air-relative velocity have already been performed Palomaki et al., as well as Gonzalez-Rocha et al.*

*Palomaki, R.T., Rose, N.T., van den Bossche, M., Sherman, T.J. and De Wekker, S.F., 2017. Wind estimation in the lower atmosphere using multirotor aircraft. Journal of Atmospheric and Oceanic Technology, 34(5), pp.1183-1191.*

*Donnell, G.W., Feight, J.A., Lannan, N. and Jacob, J.D., 2018. Wind characterization using onboard IMU of sUAS. In 2018 Atmospheric Flight Mechanics Conference (p. 2986).*

*González-Rocha, J., Woolsey, C.A., Sultan, C. and De Wekker, S.F., 2019. Sensing wind from quadrotor motion. Journal of Guidance, Control, and Dynamics, 42(4), pp.836-852.*

*Abichandani, P., Lobo, D., Ford, G., Bucci, D. and Kam, M., 2020. Wind measurement and simulation techniques in multi-rotor small unmanned aerial vehicles. IEEE Access, 8, pp.54910-54927.*

We describe in detail the calibration procedure and the post-processing analysis, proving that our method could also handle atmospheric wind well (we could estimate the atmospheric wind that matches the meteorological re-analysis). Our calibration strategy maps all the possible flight speeds of the UAS. It guarantees a comparable amount of data points for each one of them.

Moreover, as Abichandani et.al 2020 mentions when talking about the tilt angle approach: "Another limitation to this approach is the requirement of wind tunnel tests to determine the drag force and tilt angle relationship. The relationship established is unique to that specific multi-rotor vehicle. A wind tunnel to perform tests may not be available to everyone, thereby limiting the usability of this technique.". Our calibration approach can be performed in few hours and even easily repeated in order to gather further datapoints.

Studies like Neumann, Bartholmai 2015, Palomaki et.al. 2017, Donnell et.al. 2018, all build up their method like the one we call the direct method. So there is a direct relation between tilt angle and wind speed (squared or not). But none of them analyze the behavior of the Drag coefficient itself. Some studies even assume the drag coefficients to be constant over a specific range of velocities (maybe a valid assumption if the range of velocities is limited). In our case, we prove the Ca to be not constant over our range of speeds. We think mapping the behavior of the drag coefficient could be useful in developing better models and algorithms like the one proposed in Jia-Ying Wang, Bing Luo, Ming Zeng and Qing-Hao Meng,A Wind Estimation Method with an Unmanned Rotorcraft for Environmental Monitoring Tasks.

**Are substantial conclusions reached?**

*The author's claim to present a technique that does not require the use of a wind tunnel or mast towers. However, the validation experiments discussed in Section 4 were performed using a sonic anemometer, a standard practice for validation sUAS wind estimates (see references below).*
*Nolan, P.J., Pinto, J., González-Rocha, J., Jensen, A., Vezzi, C.N., Bailey, S.C., De Boer, G., Diehl, C., Laurence, R., Powers, C.W. and Foroutan, H., 2018. Coordinated unmanned aircraft system (UAS) and ground-based weather measurements to predict Lagrangian coherent structures (LCSs). Sensors, 18(12), p.4448.*
*Barbieri, L., Kral, S.T., Bailey, S.C., Frazier, A.E., Jacob, J.D., Reuder, J., Brus, D., Chilson, P.B., Crick, C., Detweiler, C. and Doddi, A., 2019. Intercomparison of small unmanned aircraft system (sUAS) measurements for atmospheric science during the LAPSE-RATE campaign. Sensors, 19(9), p.2179.*

Every new measurement technique needs to be validated; a known, reliable and precise measurement technique should be used.

In this manuscript, we first describe our measurement technique (which works without using a sonic anemometer) and then we validate the measurement technique using a sonic anemometer.

Therefore, the measurement technique does not require any additional instruments apart from the copter, but, since the goal of this paper is also to validate this method, it has to be compared to a reliable measurement that works independently from the UAS measurement.

**Are the scientific methods and assumptions valid and clearly outlined?**
*In addition to developing a model-based wind estimation technique, the authors propose simplifying the aerodynamic characteristics of sUAS by enclosing the airframe and electronic components using a Styrofoam sphere. The authors implicitly assume the airframe drag effects to be significant. However, data that support this assumption are have not been presented. On the other hand, a previous study by Powers et al. has shown multirotor sUAS drag effects to be dominated by the propeller and airflow interactions instead of airframe shape. Moreover, quadrotor experiments performed by González-Rocha et al. show the tilt variations as a function to sideslip angle to be within the noise of the measurement at different ground speeds.*
*Powers, C., Mellinger, D., Kushleyev, A., Kothmann, B. and Kumar, V., 2013. Influence of aerodynamics and proximity effects in quadrotor flight. In Experimental robotics (pp. 289-302). Springer, Heidelberg.*
*González-Rocha, J., Woolsey, C.A., Sultan, C. and De Wekker, S.F., 2019. Sensing wind from quadrotor motion. Journal of Guidance, Control, and Dynamics, 42(4), pp.836-852.*

We recently managed to compare the tilt angle output of our two DJI systems mounting the dome on one while leaving the frame exposed on the other. We added some weights on the second one in order to simulate the weight of the dome so that the comparison is meaningful. The results are reported in Figure 12a of the new version of the manuscript and the description of the experiment is described in section 5.1

However we think that what stated about the symmetry and the more regular shape provided by the encasing is anyway valid. We do not want to argue what found by Neumann, Bartholmai 2015 and Gonzálex-Rocha 2019, but rather, considering their results, the spherical cover could only grant more uniformity in the system response at different sideslip angles. We performed another test to prove this, and the result is shown in Figure 12b. Again the description of the test is in Sect 5.1.

Bot not only that, the sphere grants the same cross sectional area (body area) when the copter is tilting. Donnell 2018 while discussing about the accuracy of the tilt angle method applied to the different system he is comparing says: "With respect to the indirect methods the Solo performs the best. The Solo is a more advanced vehicle with a newer generation Pixhawk flight controller. The airframe of the Iris is larger and contains more planer surfaces with sharp edges whereas the Solo has contour features and smooth edges. The Iris's motor and body layout are symmetric only about its roll axis, excluding small features such as antenna and mini USB port on the port side of the vehicle. The Solo's airframe body is symmetric about its roll and pitch axis, excluding the undercarriage where on the forward end is a flat with a hole to mount a payload such as a camera. The combination of these factors directly influences the behavior and orientation of the vehicle while it maintains position with an incoming wind velocity". It seems then reasonable to enclose all the sharp edges and electronics cables inside a more uniform and smooth surface.

**Are the results sufficient to support the interpretations and conclusions?**
*The authors need to perform experiments to compare the inflow angle of nominal and spherical sUAS configurations over a range of ground speeds and sideslip angles.*

See section 5.1 and Fig 12.

**Is the description of experiments and calculations sufficiently complete and precise to allow their reproduction by fellow scientists (traceability of results)?**
*Yes, the description of experiments and calculations are in general complete. However, there are formulae that need to be improved for correctness.*

We updated the formulae as specified in the following comments.

**Do the authors give proper credit to related work and clearly indicate their own new/original contribution?**
*Tilt models to estimate wind velocity have been proposed before. It was difficult to understand how the work presented in this manuscript improves upon previous models.*

The calibration procedure has been explained extensively, and tested by proving that it can handle and even identify the magnitude of possible atmospheric wind. A model for the extended drag coefficient is presented and compared to the more common velocity vs tilt model highlighting advantages and disadvantages.

**Does the title clearly reflect the contents of the paper?**

*No, the wind estimation algorithm being presented is not a stand-alone technique. The implementation of this algorithm requires calibration experiments next to a conventional wind sensor.*

We use a sonic anemometer in the moment we want to validate our calibration procedure.

**Does the abstract provide a concise and complete summary?**
*The abstract does not provide a concise and complete summary of the work presented. It was difficult to appreciate what the authors*

The abstract has been partially modified in order to try to give the reader a better summary of the manuscript.

**Is the overall presentation well structured and clear?**
*The presentation of the manuscript is well structured. However, there are sections of the manuscript that need to be improved for clarity and conciseness.*

The manuscript has now been modified taking into account all the three reviewers comments. Some unclear sentences have been rephrased and some equations modified.

**Is the language fluent and precise?**
*The authors can significantly improve the language to be more precise.*

See previous comment

**Are mathematical formulae, symbols, abbreviations, and units correctly defined and used?**
*The formulae need to be revised. For example, in Eqs. (1) and (2) the rotation matrices need to be defined.*
*Additionally, the transformation presented in Eq (1) need to be transposed for correctness. Moreover, the tilt angle in Eq (3) can be estimated using the product rule.*

We included the extended expression for the three rotation matrices.

**Should any parts of the paper (text, formulae, figures, tables) be clarified, reduced, combined, or eliminated?**
*The abstract language needs to be clarified. As it stands, it is not evident that the authors are proposing a method based on flight transects for characterizing a wind estimation tilt model instead of hovering inside of a wind tunnel or next to a sonic anemometer.*

The abstract has been partially modified in order to try to give the reader a better summary of the manuscript.

**Are the number and quality of references appropriate?**

*The manuscript does not present a comprehensive survey of model-based estimation techniques.*

The introduction has been extended and a short section on on-board sensors has been added in order to make the survey of wind measurement techniques more complete.

**Is the amount and quality of supplementary material appropriate?**

*Yes.*